# Automatic Differentiation of Optimization Algorithms with Time-Varying Updates

**Sheheryar Mehmood** [1]   **Peter Ochs** [1]

## Abstract

Numerous optimization algorithms have a time-varying update rule thanks to, for instance, a changing step size, momentum parameter or, Hessian approximation. Often, such algorithms are used as solvers for the lower-level problem in bilevel optimization, and are unrolled when computing the gradient of the upper-level objective. In this paper, we apply unrolled or automatic differentiation to a time-varying iterative process and provide convergence (rate) guarantees for the resulting derivative iterates. We then adapt these convergence results and apply them to proximal gradient descent with variable step size and FISTA when solving partly-smooth problems. We test the convergence (rates) of these algorithms numerically through several experiments. Our theoretical and numerical results show that the convergence rate of the algorithm is reflected in its derivative iterates.

## 1. Introduction

For some parameter $\boldsymbol{u} \in \mathcal{U}$, we consider the following parametric iterative process

$$\boldsymbol{x}^{(k+1)}(\boldsymbol{u}) \coloneqq \mathcal{A}(\boldsymbol{x}^{(k)}(\boldsymbol{u}), \boldsymbol{u}), \qquad (\mathcal{R})$$

for $k \geq 0$, where $\boldsymbol{x}^{(0)} \in \mathcal{X}$ is the initial iterate and $\mathcal{A} \colon \mathcal{X} \times \mathcal{U} \to \mathcal{X}$ is the update mapping. The iterates $\boldsymbol{x}^{(k)}(\boldsymbol{u})$ generated by $(\mathcal{R})$ depend on $\boldsymbol{u}$ due to the dependence of $\mathcal{A}$ on $\boldsymbol{u}$. The goal of performing the iterations $(\mathcal{R})$ is mainly to solve the non-linear fixed-point equation

$$\boldsymbol{x} = \mathcal{A}(\boldsymbol{x}, \boldsymbol{u}), \qquad (\mathcal{R}_{\mathrm{e}})$$

with respect to $\boldsymbol{x}$ for each parameter $\boldsymbol{u}$. A simple example is gradient descent with appropriate step size $\alpha > 0$

[1] Department of Mathematics & Computer Science, Saarland University, Saarbrücken, Germany. Correspondence to: Sheheryar Mehmood <mehmood@math.uni-sb.de>.

*Proceedings of the 42$^{nd}$ International Conference on Machine Learning*, Vancouver, Canada. PMLR 267, 2025. Copyright 2025 by the author(s).

where we define $\mathcal{A}(\boldsymbol{x}, \boldsymbol{u}) \coloneqq \boldsymbol{x} - \alpha \nabla_{\boldsymbol{x}} F(\boldsymbol{x}, \boldsymbol{u})$ for solving a parametric optimization problem of the form

$$\min_{\boldsymbol{x} \in \mathcal{X}} \; F(\boldsymbol{x}, \boldsymbol{u}), \qquad (\mathcal{P})$$

with a smooth objective $F \colon \mathcal{X} \times \mathcal{U} \to \mathbb{R}$. In this case $(\mathcal{R}_{\mathrm{e}})$ reduces to the optimality condition $\nabla_{\boldsymbol{x}} F(\boldsymbol{x}, \boldsymbol{u}) = 0$.

There is a wide range of practical applications in which, problems of type $(\mathcal{R}_{\mathrm{e}})$ or $(\mathcal{P})$ appear as a constraint within a minimization problem with respect to the parameter $\boldsymbol{u}$. Usually, for some smooth $\ell \colon \mathcal{X} \times \mathcal{U} \to \mathbb{R}$, this can be compactly stated as

$$\min_{\boldsymbol{u} \in \mathcal{U}} \; \ell(\psi(\boldsymbol{u}), \boldsymbol{u}), \qquad (1)$$

where, given any $\boldsymbol{u} \in \mathcal{U}$, $\psi(\boldsymbol{u})$ is a solution of $(\mathcal{R}_{\mathrm{e}})$ or $(\mathcal{P})$. This kind of optimization, formally known as bilevel optimization (Dempe et al., 2015; Dempe & Zemkoho, 2020) has emerged as a crucial tool in machine learning, playing a foundational role in various applications such as hyperparameter optimization (Bengio, 2000; Domke, 2012; Maclaurin et al., 2015), implicit neural networks (Amos & Kolter, 2017; Agrawal et al., 2019; Bai et al., 2019; 2020; Winston & Kolter, 2020), meta-learning (Hospedales et al., 2021), and neural architecture search (Liu et al., 2019).

Solving (1) through gradient-based methods requires computing the gradient of the upper objective $\ell(\psi(\boldsymbol{u}), \boldsymbol{u})$ with respect to $\boldsymbol{u}$ which, in turn, requires computing the derivative of the solution mapping $\boldsymbol{u} \mapsto \psi(\boldsymbol{u})$, that is, $D\psi(\boldsymbol{u})$ by using the chain rule. When $\ell$ and $F$ (or $\mathcal{A}$) satisfy certain smoothness and regularity assumptions, this can be achieved by two different methods. (i) One is known as *Implicit Differentiation* (ID) which leverages the classical Implicit Function Theorem or IFT (Dontchev & Rockafellar, 2009, Theorem 1B.1) applied to $(\mathcal{R}_{\mathrm{e}})$ to estimate

$$\begin{aligned} D\psi(\boldsymbol{u}) &= (I - D_{\boldsymbol{x}}\mathcal{A}(\boldsymbol{x}, \boldsymbol{u}))^{-1} D_{\boldsymbol{u}}\mathcal{A}(\boldsymbol{x}, \boldsymbol{u}) \\ &= -\nabla_{\boldsymbol{x}}^2 F(\boldsymbol{x}, \boldsymbol{u})^{-1} D_{\boldsymbol{u}} \nabla_x F(\boldsymbol{x}, \boldsymbol{u}), \end{aligned} \qquad (2)$$

where $\boldsymbol{x} \coloneqq \psi(\boldsymbol{u})$ (see Theorem A.10 for more details). The key assumption for IFT is the strict bound on the spectral radius $\rho(D_{\boldsymbol{x}}\mathcal{A}(\psi(\boldsymbol{u}), \boldsymbol{u})) < 1$, which we call the contraction property since it is a sufficient condition for $\mathcal{A}$

to be 1-Lipschitz or contractive. A direct consequence is the linear convergence of $\boldsymbol{x}^{(k)}(\boldsymbol{u})$ to $\psi(\boldsymbol{u})$ (Polyak, 1987, Section 2.1.2, Theorem 1). The output of ID (2) depends on accurately solving ($\mathcal{R}_e$) — possibly through ($\mathcal{R}$) — and solving the linear system efficiently. (ii) The other technique is known as *Unrolled, Iterative or Automatic Differentiation* (AD) or simply *Unrolling* where $D\psi(\boldsymbol{u})$ is estimated by applying automatic differentiation (Wengert, 1964; Linnainmaa, 1970; Griewank & Walther, 2008) to the output $\boldsymbol{x}^{(K)}(\boldsymbol{u})$ of ($\mathcal{R}$) after $K \in \mathbb{N}$ iterations. The update rule for forward mode AD applied to $\boldsymbol{x}^{(K)}$ is given by

$$
\begin{aligned}
D\boldsymbol{x}^{(k+1)}(\boldsymbol{u}) &\coloneqq D_{\boldsymbol{x}}\mathcal{A}(\boldsymbol{x}^{(k)}(\boldsymbol{u}),\boldsymbol{u})D\boldsymbol{x}^{(k)}(\boldsymbol{u}) \\
&\quad + D_{\boldsymbol{u}}\mathcal{A}(\boldsymbol{x}^{(k)}(\boldsymbol{u}),\boldsymbol{u}),
\end{aligned} \quad (\mathcal{DR})
$$

for $0 \le k < K$ where we may assume $D\boldsymbol{x}^{(0)}(\boldsymbol{u}) = 0$. The effective use of AD relies on the guarantee that $D\boldsymbol{x}^{(K)}(\boldsymbol{u})$ accurately estimates $D\psi(\boldsymbol{u})$ for a preferably small $K$. More formally, we require the convergence of $D\boldsymbol{x}^{(K)}(\boldsymbol{u})$ to $D\psi(\boldsymbol{u})$ as $K \to \infty$, with a convergence rate comparable to that of $\boldsymbol{x}^{(k)}(\boldsymbol{u})$.

The reverse mode AD or Backpropagation (Rumelhart et al., 1986; Baydin et al., 2018) has a different update rule, however, the result computed is the same in the end (Griewank & Walther, 2008). Due to the nature of the iterations ($\mathcal{DR}$), we will use $\boldsymbol{x}^{(K)}$ and $\boldsymbol{x}^{(k)}$ interchangeably. The reverse mode AD has a memory overhead since it requires storing the iterates $(\boldsymbol{x}^{(k)})_{0 \le k < K}$. Even though this is in contrast with ID which only depends on $\boldsymbol{x}^{(K)}$, AD is still a popular choice for estimating $D\psi(\boldsymbol{u})$. A crucial advantage of AD is that it provides a nice blackbox implementation thanks to the powerful autograd libraries included in PyTorch (Paszke et al., 2019), TensorFlow (Abadi et al., 2016), and JAX (Bradbury et al., 2018) whereas ID requires either a custom implementation (Bolte et al., 2023, see Remark 2) or using special libraries (Blondel et al., 2022; Ren et al., 2023, etc.).

### 1.1. Convergence of Automatic Differentiation

The convergence of the derivative sequence $D\boldsymbol{x}^{(k)}(\boldsymbol{u})$ for AD of ($\mathcal{R}$) were studied classically for linear (Fischer, 1991) and general iteration maps (Gilbert, 1992). More recently, the convergence analysis was provided for gradient descent (Lorraine et al., 2020; Mehmood & Ochs, 2020), the Heavyball method (Mehmood & Ochs, 2020), proximal gradient descent for lasso-like problems (Bertrand et al., 2022), the Sinkhorn-Knopp algorithm (Pauwels & Vaiter, 2023), the super-linearly convergent algorithms (Bolte et al., 2023), and stochastic gradient descent (Iutzeler et al., 2024). The convergence results were extended to non-smooth iterative processes by Bolte et al. (2022) using conservative calculus (Bolte & Pauwels, 2020; 2021). Ablin et al. (2020) theoretically established the better convergence rate for computing the gradient of the value function through AD and ID. Usu-

ally, the proof of convergence of $D\boldsymbol{x}^{(k)}(\boldsymbol{u})$ relies on the contraction property in each of the above works with an exception of Pauwels & Vaiter (2023). The linear rate of convergence requires (local) Lipschitz continuity of $D\mathcal{A}$. A slightly different line of study on the problematic behaviour of automatic differentiation in the earlier iterations was carried out by Scieur et al. (2022).

### 1.2. Iteration-Dependent Updates

Most iterative algorithms for solving ($\mathcal{P}$) which include gradient descent with line search, Nesterov's accelerated gradient (Nesterov, 1983), and Quasi-Newton methods (Davidon, 1959; Fletcher & Powell, 1963; Broyden, 1970; Fletcher, 1970, etc.) cannot be modelled by ($\mathcal{R}$). This happens because the update mapping is changing at every iteration; $\mathcal{A}$ in ($\mathcal{R}$) is replaced by $\mathcal{A}_k\colon \mathcal{X} \times \mathcal{U} \to \mathcal{X}$ giving us the update

$$
\boldsymbol{x}^{(k+1)}(\boldsymbol{u}) \coloneqq \mathcal{A}_k(\boldsymbol{x}^{(k)}(\boldsymbol{u}),\boldsymbol{u}). \quad (\mathcal{R}_k)
$$

For example, for gradient descent with line search, $\mathcal{A}_k(\boldsymbol{x},\boldsymbol{u}) \coloneqq \boldsymbol{x} - \alpha_k \nabla_{\boldsymbol{x}} F(\boldsymbol{x},\boldsymbol{u})$. When $\mathcal{A}_k$ is $C^1$-smooth for all $k \in \mathbb{N}$, the modified update rule for performing forward mode AD on ($\mathcal{R}_k$) reads:

$$
\begin{aligned}
D\boldsymbol{x}^{(k+1)}(\boldsymbol{u}) &\coloneqq D_{\boldsymbol{x}}\mathcal{A}_k(\boldsymbol{x}^{(k)}(\boldsymbol{u}),\boldsymbol{u})D\boldsymbol{x}^{(k)}(\boldsymbol{u}) \\
&\quad + D_{\boldsymbol{u}}\mathcal{A}_k(\boldsymbol{x}^{(k)}(\boldsymbol{u}),\boldsymbol{u}).
\end{aligned} \quad (\mathcal{DR}_k)
$$

Beck (1994) studied this problem for the $C^1$-smooth sequence of functions $\mathcal{A}_k$ with uniformly bounded derivatives and $D\mathcal{A}_k$ converging to $D\mathcal{A}$ pointwise. They showed that for such a sequence, when $(\boldsymbol{x}^{(k)}(\boldsymbol{u}))_{k \in \mathbb{N}}$ is generated by ($\mathcal{R}_k$) to solve the system ($\mathcal{R}_e$), then $(D\boldsymbol{x}^{(k)}(\boldsymbol{u}))_{k \in \mathbb{N}}$ generated by ($\mathcal{DR}_k$) converges to $D\psi(\boldsymbol{u})$ defined in (2). *However, their results cannot account for the linear convergence of $D\boldsymbol{x}^{(k)}(\boldsymbol{u})$ in their given setting (see Section A.3). Moreover, their setting excludes the more general cases where $\mathcal{A}_k$ does not converge, for instance, gradient descent with line search.*

Griewank et al. (1993) studied the special setting for solving the non-linear system $G(\boldsymbol{x},\boldsymbol{u}) = 0$ with respect to $\boldsymbol{x}$ through preconditioned Picard iterations, that is, $\mathcal{A}_k(\boldsymbol{x},\boldsymbol{u}) \coloneqq \boldsymbol{x} - P_k(\boldsymbol{u})G(\boldsymbol{x},\boldsymbol{u})$. The parameter-dependent preconditioner $P_k\colon \mathcal{U} \to \mathcal{L}(\mathcal{X},\mathcal{X})$ is changing at every iteration and may not converge. They demonstrated linear convergence of $D\boldsymbol{x}^{(k)}(\boldsymbol{u})$ for a $C^1$-smooth $G\colon \mathcal{X} \times \mathcal{U} \to \mathcal{X}$ with bounded maps $D_{\boldsymbol{x}}G$ and $(\boldsymbol{x},\boldsymbol{u}) \mapsto D_{\boldsymbol{x}}G(\boldsymbol{x},\boldsymbol{u})^{-1}$ and Lipschitz continuous map $DG$. The map $P_k$ is assumed to be $C^1$-smooth with bounded (potentially 0) derivative for every $k \in \mathbb{N}$.

Riis (2020); Mehmood & Ochs (2022) considered applying AD when $\mathcal{A}_k$ denotes the update mapping of FISTA (Beck & Teboulle, 2009) (which we will call Accelerated Proximal Gradient or APG) applied to ($\mathcal{P}$) for non-smooth

$F := f + g$ with step size $\alpha_k$ and extrapolation parameter $\beta_k$. They adapted the results of Beck (1994) and demonstrated the convergence of $D\boldsymbol{x}^{(k)}(\boldsymbol{u})$ to $D\psi(\boldsymbol{u})$ when $f$ is $C^2$-smooth with a Lipschitz continuous gradient, $g$ is partly smooth (Lewis, 2002), the sequences $(\alpha_k)_{k\in\mathbb{N}}$ and $(\beta_k)_{k\in\mathbb{N}}$ converge and $F$ additionally satisfies restricted injectivity and non-degeneracy conditions (see, for example, Lewis, 2002). $D\psi(\boldsymbol{u})$ is computed by applying the Implicit Function Theorem for partly smooth functions (Lewis, 2002, Theorem 5.7). Partly smooth functions are a class of special non-smooth functions which exhibit smoothness when restricted to a low-dimensional manifold $\mathcal{M}$ and model various practical regularization functions (Vaiter et al., 2017). The non-smooth or sharp behaviour occurs only when we move orthogonal to $\mathcal{M}$. The combination of restricted injectivity and non-degeneracy assumptions induces the contraction property in proximal gradient descent or PGD (Liang et al., 2014) and APG (Liang et al., 2017) near the solution. *The results of Riis (2020); Mehmood & Ochs (2022) do not explain the linear convergence of AD of APG or even the convergence of AD of PGD with non-converging variable step size.*

Iutzeler et al. (2024) tackle the problem of applying AD on Stochastic Gradient Descent or SGD and ensure convergence of the derivative sequence by considering the derivative recursion as a so-called "inexact SGD" applied to solve a stochastic quadratic problem whose solution is the desired derivative of the solution mapping. Although, our deterministic and their stochastic settings both work with time-varying algorithms, we can not simply deduce one from the other in a trivial manner.

### 1.3. Contributions

In this paper, we aim to resolve the theoretical gaps presented in Section 1.2 and therefore reinforce the usefulness of AD. In particular, the main contributions of this paper are the following:

(i) We strengthen the results of Beck (1994) by providing a convergence rate analysis for $D\boldsymbol{x}^{(k)}(\boldsymbol{u})$ generated by $(\mathcal{DR}_k)$ after equipping their setting with an additional assumption that is satisfied by most optimization algorithms (see Assumption 2.1(ii) and Remark 2.1).

(ii) We establish (linear) convergence of $D\boldsymbol{x}^{(k)}(\boldsymbol{u})$ beyond the setting in (i) without the pointwise convergence condition of $\mathcal{A}_k$ to $\mathcal{A}$ (see Assumption 2.2 and 2.3).

(iii) We demonstrate the convergence of $D\boldsymbol{x}^{(k)}(\boldsymbol{u})$ for Proximal Gradient Descent (Lions & Mercier, 1979) or PGD with variable step size and Accelerated Proximal Gradient (Beck & Teboulle, 2009) or APG by adapting the results for (ii) and (i) respectively. In contrast to Riis (2020) and Mehmood & Ochs (2022), which rely

on the framework of Beck (1994), our results provide *convergence and convergence rate guarantees* for PGD with variable step size and *convergence rate guarantees* for APG. We show that the rate of convergence of $\boldsymbol{x}^{(k)}(\boldsymbol{u})$ is reflected in that of $D\boldsymbol{x}^{(k)}(\boldsymbol{u})$ for both algorithms.

### 1.4. Notation

In this paper, $\mathcal{X}$ and $\mathcal{U}$ are Euclidean spaces and $\overline{\mathbb{R}} := [-\infty, \infty]$. With a slight abuse of notation, $\|\cdot\|$ denotes any norm on each of the spaces $\mathcal{X}$ and $\mathcal{U}$ including the one induced by the inner product $\langle\cdot,\cdot\rangle$. The space of linear operators from $\mathcal{U}$ to $\mathcal{X}$ is denoted by $\mathcal{L}(\mathcal{U},\mathcal{X})$. Any operator norm on the spaces $\mathcal{L}(\mathcal{U},\mathcal{X})$ and $\mathcal{L}(\mathcal{X},\mathcal{U})$ is also denoted by $\|\cdot\|$. We say that $\boldsymbol{w}^{(k)} = \mathrm{o}(\boldsymbol{z}^{(k)})$ iff $\|\boldsymbol{w}^{(k)}\| = \mathrm{o}(\|\boldsymbol{z}^{(k)}\|)$ and $\boldsymbol{w}^{(k)} = \mathcal{O}(\boldsymbol{z}^{(k)})$ iff $\|\boldsymbol{w}^{(k)}\| = \mathcal{O}(\|\boldsymbol{z}^{(k)}\|)$.

The affine hull of a set $C \subset \mathcal{X}$ is denote by $\mathrm{aff}\, C$ whereas $\mathrm{par}\, C = \mathrm{aff}\, C - \mathrm{aff}\, C$ is a linear subspace parallel to $C$. The relative interior of a convex set $C$ (Rockafellar, 1970, p. 44) is denoted by $\mathrm{ri}\, C$ and is always non-empty whenever $C$ is non-empty (Rockafellar, 1970, Theorem 6.2). For any $\alpha > 0$ and proper and lower semi-continuous $g: \mathcal{X} \times U \to \overline{\mathbb{R}}$ where $g(\cdot, \boldsymbol{u})$ is convex for all $\boldsymbol{u} \in U \subset \mathcal{U}$, we define $\mathrm{P}_{\alpha g}: \mathcal{X} \times U \to \mathcal{X}$ by $\mathrm{P}_{\alpha g}(\cdot, \boldsymbol{u}) = \mathrm{prox}_{\alpha g(\cdot,\boldsymbol{u})}$, that is,

$$\mathrm{P}_{\alpha g}(\boldsymbol{w}, \boldsymbol{u}) := \underset{\boldsymbol{x}\in\mathcal{X}}{\mathrm{argmin}}\, \alpha g(\boldsymbol{x}, \boldsymbol{u}) + \frac{1}{2}\|\boldsymbol{x} - \boldsymbol{w}\|^2. \quad (3)$$

In this paper we use the notion of regular subdifferential (Rockafellar & Wets, 1998, Definition 8.3) of $g$. When $U$ is open and for any $\boldsymbol{x} \in \mathcal{X}$, $g(\boldsymbol{x}, \cdot)$ is differentiable at $\boldsymbol{u} \in U$, we obtain $\partial g(\boldsymbol{x}, \boldsymbol{u}) = \partial_{\boldsymbol{x}} g(\boldsymbol{x}, \boldsymbol{u}) \times \{\nabla_{\boldsymbol{u}} g(\boldsymbol{x}, \boldsymbol{u})\}$ (Mehmood & Ochs, 2022, Lemma 1).

We use standard notions of Riemannian Geometry, which we define in Section A.4 in the appendix. We refer a $C^k$-smooth $m$-dimensional submanifold by simply $C^k$-smooth manifold. Furthermore, the natural embedding of $\mathcal{M}$ in $\mathcal{X}$ allows us to define a Riemannian metric on $\mathcal{M}$, making it a Riemannian manifold. For any $\boldsymbol{x} \in \mathcal{M}$, the map $\Pi: \mathcal{M} \to \mathcal{L}(\mathcal{X}, \mathcal{X})$ outputs the orthogonal projection onto the tangent space of $\mathcal{M}$ at $\boldsymbol{x}$, that is, $\Pi(\boldsymbol{x}) := \mathrm{proj}_{T_{\boldsymbol{x}}\mathcal{M}}$ while $\Pi^\perp: \mathcal{M} \to \mathcal{L}(\mathcal{X}, \mathcal{X})$ gives us the orthogonal projection onto the normal space of $\mathcal{M}$ at $\boldsymbol{x}$, that is, $\Pi^\perp(\boldsymbol{x}) := \mathrm{proj}_{N_{\boldsymbol{x}}\mathcal{M}}$. For a function $g: \mathcal{X} \times \mathcal{U} \to \overline{\mathbb{R}}$ with values $g(\boldsymbol{x}, \boldsymbol{u})$ which is $C^2$-smooth when restricted to $\mathcal{M} \times U$ where $\mathcal{M} \subset \mathcal{X}$ is a $C^2$-smooth manifold and $U \subset \mathcal{U}$ is an open set, we respectively denote by $\nabla_{\mathcal{M}} g$ and $\nabla_{\mathcal{M}}^2 g$, the Riemannian gradient and Hessian of $g$ relative to $\mathcal{M}$ with respect to $\boldsymbol{x}$. We follow Mehmood & Ochs (2022, Remark 22(iv)) to express the Riemannian gradient $\nabla_{\mathcal{M}\times U} g$ and Hessian $\nabla_{\mathcal{M}\times U}^2 g$ of $g$.

# 2. Differentiation of Iteration-Dependent Algorithms

In this section, we lay down the assumptions on the sequence $(\mathcal{A}_k)_{k \in \mathbb{N}}$ and provide convergence and convergence rate guarantees for the derivative iterations defined in ($\mathcal{DR}_k$). In particular we establish (i) convergence rates when $\mathcal{A}_k$ has a pointwise limit $\mathcal{A}$, and (ii) convergence and convergence rates when $\mathcal{A}_k$ does not have a pointwise limit. For a better comparison and understanding, this section is adapted to follow the same pattern as Section A.3, which summarizes the results by Beck (1994).

## 2.1. Problem Setting

For a given $\boldsymbol{u}^*$, we assume that $\boldsymbol{x}^*$ is the solution that we are trying to estimate through ($\mathcal{R}_k$). Because the pointwise limit $\mathcal{A}$ of $\mathcal{A}_k$ may not exist, we assume that $\boldsymbol{x}^*$ is a fixed point of $\mathcal{A}_k(\cdot, \boldsymbol{u}^*)$ for every $k \in \mathbb{N}$. This seemingly restrictive assumption is satisfied by the optimization algorithms highlighted in Section 1.2. The application of the Implicit Function Theorem on this sequence of fixed-point equations requires the contraction property, that is, the strict bound on some operator norm $\|D_{\boldsymbol{x}} \mathcal{A}_k(\boldsymbol{x}^*, \boldsymbol{u}^*)\| < 1$ eventually for all $k \in \mathbb{N}$. Furthermore, we assume that the sequence of affine maps $(X \mapsto D_{\boldsymbol{x}} \mathcal{A}_k(\boldsymbol{x}^*, \boldsymbol{u}^*)X + D_{\boldsymbol{u}} \mathcal{A}_k(\boldsymbol{x}^*, \boldsymbol{u}^*))_{k \in \mathbb{N}}$ share a common fixed-point $X_* \in \mathcal{L}(\mathcal{X}, \mathcal{X})$ — the reason will become clear shortly. Finally, to show the convergence of the derivative sequence $(D\boldsymbol{x}^{(k)}(\boldsymbol{u}^*))_{k \in \mathbb{N}}$, we assume that the sequence $(\boldsymbol{x}^{(k)}(\boldsymbol{u}^*))_{k \in \mathbb{N}}$ converges to $\boldsymbol{x}^*$. More formally, given $(\boldsymbol{x}^{(0)}, \boldsymbol{x}^*, \boldsymbol{u}^*, X_*) \in \mathcal{X} \times \mathcal{X} \times \mathcal{U} \times \mathcal{L}(\mathcal{X}, \mathcal{X})$, $V \in \mathcal{N}_{(\boldsymbol{x}^*, \boldsymbol{u}^*)}$, $(\mathcal{A}_k)_{k \in \mathbb{N}_0}$, and a norm $\| \cdot \|$ on $\mathcal{L}(\mathcal{X}, \mathcal{X})$ induced by some vector norm, we assume that the following conditions hold.

**Assumption 2.1.** (i) $\mathcal{A}_k|_V$ is $C^1$-smooth for all $k$,

(ii) $\boldsymbol{x}^* = \mathcal{A}_k(\boldsymbol{x}^*, \boldsymbol{u}^*)$ and $X_* = D_{\boldsymbol{x}} \mathcal{A}_k(\boldsymbol{x}^*, \boldsymbol{u}^*)X_* + D_{\boldsymbol{u}} \mathcal{A}_k(\boldsymbol{x}^*, \boldsymbol{u}^*)$ for all $k$,

(iii) $\limsup_{k \to \infty} \|D_{\boldsymbol{x}} \mathcal{A}_k(\boldsymbol{x}^*, \boldsymbol{u}^*)\| < 1$, and

(iv) $\boldsymbol{x}^{(k)}(\boldsymbol{u}^*)$ generated by ($\mathcal{R}_k$) has limit $\boldsymbol{x}^*$ such that $(\boldsymbol{x}^{(k)}(\boldsymbol{u}^*), \boldsymbol{u}^*) \in V$ for all $k$.

*Remark* 2.1. (i) Unlike Assumption A.1 from Beck (1994), where the existence of $\mathcal{A}$ gives us both the fixed-point and its derivative at $\boldsymbol{u}^*$ (see A.1(ii)), our Assumption 2.1 does not require the existence of such a limit which allows us to work directly with the family $(\mathcal{A}_k)_{k \in \mathbb{N}}$, each of which provides the solution and its derivative at $\boldsymbol{u}^*$ (see 2.1(ii)). Moreover, the spectral radius condition is imposed on the $\limsup$ of the spectral radii of derivative of $\mathcal{A}_k$ (see 2.1(iii)) rather than on the derivative of $\mathcal{A}$ (see A.1(iii)).

(ii) A special case of Assumption 2.1(iii) arises when for some $C^1$-smooth mapping $\mathcal{A}: V \to \mathcal{X}$, we have

$\boldsymbol{x}^* = \mathcal{A}(\boldsymbol{x}^*, \boldsymbol{u}^*)$, $D\mathcal{A}_k(\boldsymbol{x}^*, \boldsymbol{u}^*) \to D\mathcal{A}(\boldsymbol{x}^*, \boldsymbol{u}^*)$ and $\rho(D_{\boldsymbol{x}} \mathcal{A}(\boldsymbol{x}^*, \boldsymbol{u}^*)) < 1$.

## 2.2. Implicit Differentiation

Combining Assumptions 2.1((i)–(iii)) allows us to invoke the Implicit Function Theorem on $\boldsymbol{x}^* = \mathcal{A}_k(\boldsymbol{x}^*, \boldsymbol{u}^*)$ for every $k$ large enough and obtain a $C^1$-smooth fixed-point map $\psi_k$ on a neighbourhood $U_k \in \mathcal{N}_{\boldsymbol{u}^*}$. In particular, we have the following result.

**Lemma 2.2.** *Let* $(\boldsymbol{x}^*, \boldsymbol{u}^*, X_*) \in \mathcal{X} \times \mathcal{U} \times \mathcal{L}(\mathcal{X}, \mathcal{X})$, $V \in \mathcal{N}_{(\boldsymbol{x}^*, \boldsymbol{u}^*)} (\mathcal{A}_k)_{k \in \mathbb{N}_0}$, *and* $\| \cdot \|$ *be such that Assumption 2.1 ((i)–(iii)) is satisfied. Then there exists* $K \in \mathbb{N}$ *such that* $\forall k \geq K$, *there exists* $U_k \in \mathcal{N}_{\boldsymbol{u}^*}$ *and a* $C^1$-*smooth* $\psi_k \colon U_k \to \mathcal{X}$ *such that* $\forall \boldsymbol{u} \in U_k$, $\psi_k(\boldsymbol{u}) = \mathcal{A}_k(\psi_k(\boldsymbol{u}), \boldsymbol{u})$, *and*

$$D\psi_k(\boldsymbol{u}) = \left(I - D_{\boldsymbol{x}} \mathcal{A}_k(\psi_k(\boldsymbol{u}), \boldsymbol{u})\right)^{-1} D_{\boldsymbol{u}} \mathcal{A}_k(\psi_k(\boldsymbol{u}), \boldsymbol{u}), \tag{4}$$

*so that* $D\psi_k(\boldsymbol{u}^*) = X_*$.

*Proof.* The proof is in Section B.1 in the appendix. $\square$

*Remark* 2.3. If we replace Assumptions 2.1(ii) and 2.1(iii) by a slightly stronger assumption, that is, "*there exists* $U \in \mathcal{N}_{\boldsymbol{u}^*}$ *such that for any* $(\boldsymbol{x}, \boldsymbol{u}) \in \mathcal{X} \times U$, $\boldsymbol{x} = \mathcal{A}_k(\boldsymbol{x}, \boldsymbol{u})$ *for some* $k \in \mathbb{N}$ *implies* $\boldsymbol{x} = \mathcal{A}_k(\boldsymbol{x}, \boldsymbol{u})$ *for every* $k \in \mathbb{N}$, *and* $\limsup_{k \to \infty} \|D_{\boldsymbol{x}} \mathcal{A}_k(\boldsymbol{x}, \boldsymbol{u})\| < 1$", we obtain a shared $C^1$-smooth fixed-point map $\psi: U \to \mathcal{X}$, such that, $U_k = U$ and $\psi_k = \psi$ for all $k \in \mathbb{N}$.

## 2.3. Automatic Differentiation

In Lemma 2.2, we just showed that under Assumption 2.1, the common fixed-point $X_*$ of the mappings $X \mapsto D_{\boldsymbol{x}} \mathcal{A}_k(\boldsymbol{x}^*, \boldsymbol{u}^*)X + D_{\boldsymbol{u}} \mathcal{A}_k(\boldsymbol{x}^*, \boldsymbol{u}^*)$ for $k \in \mathbb{N}$ represents the derivative of the solution at $\boldsymbol{u} = \boldsymbol{u}^*$. We now show that the derivative sequence generated by ($\mathcal{DR}_k$) for $\boldsymbol{u} = \boldsymbol{u}^*$ converges to $X_*$. Instead of assuming a pointwise limit for $D\mathcal{A}_k$, we assert that $D\mathcal{A}_k(\boldsymbol{x}^{(k)}(\boldsymbol{u}^*), \boldsymbol{u}^*) - D\mathcal{A}_k(\boldsymbol{x}^*, \boldsymbol{u}^*)$ tends to $0$.

**Assumption 2.2.** The sequence $(D\mathcal{A}_k(\boldsymbol{x}^{(k)}(\boldsymbol{u}^*), \boldsymbol{u}^*) - D\mathcal{A}_k(\boldsymbol{x}^*, \boldsymbol{u}^*))_{k \in \mathbb{N}_0}$ converges to $0$.

*Remark* 2.4. (i) In Assumption A.2 of Beck (1994), where $\mathcal{A}$ exists, the sequence $(D\mathcal{A}_k(\boldsymbol{x}^{(k)}(\boldsymbol{u}^*), \boldsymbol{u}^*) - D\mathcal{A}(\boldsymbol{x}^*, \boldsymbol{u}^*))_{k \in \mathbb{N}_0}$ is assumed to converge to $0$. Rewriting

$$\begin{aligned} D\mathcal{A}_k(\boldsymbol{x}^{(k)}(\boldsymbol{u}^*), \boldsymbol{u}^*) &- D\mathcal{A}(\boldsymbol{x}^*, \boldsymbol{u}^*) \\ &= D\mathcal{A}_k(\boldsymbol{x}^{(k)}(\boldsymbol{u}^*), \boldsymbol{u}^*) - D\mathcal{A}_k(\boldsymbol{x}^*, \boldsymbol{u}^*) \quad (5) \\ &+ D\mathcal{A}_k(\boldsymbol{x}^*, \boldsymbol{u}^*) - D\mathcal{A}(\boldsymbol{x}^*, \boldsymbol{u}^*). \end{aligned}$$

suggests that this convergence follows, for instance, when our Assumption 2.2 holds and $D\mathcal{A}_k$ converges

pointwise to $D\mathcal{A}$ at $(\boldsymbol{x}^*, \boldsymbol{u}^*)$. We demonstrate this difference for Gradient Descent with variable step size in Example A.2.

(ii) Combining Assumption 2.2 with the special case of Remark 2.1, we obtain the setting of Assumption A.2 (see Remark 2.4(i)).

(iii) A sufficient condition for Assumption 2.2 is equicontinuity (see Section A.1 for definition and an example) of $(\mathcal{A}_k)_{k \in \mathbb{N}_0}$ at $(\boldsymbol{x}^*, \boldsymbol{u}^*)$.

We are now ready to establish the convergence of the derivative sequence $(D\boldsymbol{x}^{(k)}(\boldsymbol{u}^*))_{k \in \mathbb{N}}$ to $X_*$ when Assumptions 2.1 and 2.2 hold.

**Theorem 2.5.** *Let $(\boldsymbol{x}^{(0)}, \boldsymbol{x}^*, \boldsymbol{u}^*) \in \mathcal{X} \times \mathcal{X} \times \mathcal{U}$, $V \in \mathcal{N}_{(\boldsymbol{x}^*, \boldsymbol{u}^*)}$, and $(\mathcal{A}_k)_{k \in \mathbb{N}_0}$ be such that Assumptions 2.1 and 2.2 are satisfied. Then the sequence $(D\boldsymbol{x}^{(k)}(\boldsymbol{u}^*))_{k \in \mathbb{N}_0}$ generated by $(\mathcal{DR}_k)$ converges to $X_*$ which is equal to $(I - D_{\boldsymbol{x}}\mathcal{A}_k(\boldsymbol{x}^*, \boldsymbol{u}^*))^{-1} D_{\boldsymbol{u}}\mathcal{A}_k(\boldsymbol{x}^*, \boldsymbol{u}^*)$ for all $k \in \mathbb{N}$ (see Assumption 2.1(ii)).*

*Proof.* The proof is in Section B.2 in the appendix. $\qquad\square$

The derivative iterates $D\boldsymbol{x}^{(k)}(\boldsymbol{u}^*)$ additionally converge with a linear rate when the sequences $\boldsymbol{x}^{(k)}(\boldsymbol{u}^*)$ and $D\mathcal{A}_k(\boldsymbol{x}^{(k)}(\boldsymbol{u}^*), \boldsymbol{u}^*) - D\mathcal{A}_k(\boldsymbol{x}^*, \boldsymbol{u}^*)$ converge linearly.

**Assumption 2.3.** The sequence $(\boldsymbol{x}^{(k)}(\boldsymbol{u}^*))_{k \in \mathbb{N}_0}$ converges linearly to $\boldsymbol{x}^*$ and

$$D\mathcal{A}_k(\boldsymbol{x}^{(k)}(\boldsymbol{u}^*), \boldsymbol{u}^*) - D\mathcal{A}_k(\boldsymbol{x}^*, \boldsymbol{u}^*) = \mathcal{O}(\boldsymbol{x}^{(k)}(\boldsymbol{u}^*) - \boldsymbol{x}^*). \tag{6}$$

*Remark* 2.6. (i) Assumptions A.3 (from the appendix) and 2.3 (ours) both require the corresponding sequences in Assumptions A.2 and 2.2, respectively, to converge linearly. From (5), we see that Assumption A.3 holds when Assumption 2.3 holds *and* $D\mathcal{A}_k(\boldsymbol{x}^*, \boldsymbol{u}^*) \to D\mathcal{A}(\boldsymbol{x}^*, \boldsymbol{u}^*)$ converges linearly (see also Example A.4). Since the latter condition is not practical in general, Assumption A.3 does not appear in Beck (1994) and is given here only for completeness.

(ii) A sufficient condition for Assumption 2.3 is the Lipschitz continuity of $(D\mathcal{A}_k)_{k \in \mathbb{N}_0}$ in $\boldsymbol{x}$ uniformly in $\boldsymbol{u}$ and $k$ (see Section A.1 for definition and an example).

**Theorem 2.7.** *Let $(\boldsymbol{x}^{(0)}, \boldsymbol{x}^*, \boldsymbol{u}^*) \in \mathcal{X} \times \mathcal{X} \times \mathcal{U}$, $V \in \mathcal{N}_{(\boldsymbol{x}^*, \boldsymbol{u}^*)}$, and $(\mathcal{A}_k)_{k \in \mathbb{N}_0}$ be such that Assumptions 2.1 and 2.3 are satisfied. Then the sequence $(D\boldsymbol{x}^{(k)}(\boldsymbol{u}^*))_{k \in \mathbb{N}_0}$ generated by $(\mathcal{DR}_k)$ converges linearly to $X_*$. In particular, for all $\delta \in (0, 1 - \rho)$, there exist $\xi_1$, $\xi_2$, and $K \in \mathbb{N}$, such that for all $k \geq K$, we have*

$$\|D\boldsymbol{x}^{(k)}(\boldsymbol{u}^*) - X_*\| \leq \xi_1(\rho + \delta)^{k-K}$$
$$+ \xi_2(k - K)\max(\rho + \delta, q_{\boldsymbol{x}})^{k-K}, \tag{7}$$

*where $\rho := \limsup_{k \to \infty} \rho(D_{\boldsymbol{x}}\mathcal{A}_k(\boldsymbol{x}^*, \boldsymbol{u}^*))$, and $q_{\boldsymbol{x}} < 1$ is the convergence rate of $(\boldsymbol{x}^{(k)}(\boldsymbol{u}^*))_{k \in \mathbb{N}_0}$.*

*Proof.* The proof is in Section B.3 in the appendix. $\qquad\square$

*Remark* 2.8. In (7), $\xi_1$ depends on the derivative error $D\boldsymbol{x}^{(K)}(\boldsymbol{u}^*) - X^*$ while $\xi_2$ depends on the derivative errors $D\boldsymbol{x}^{(K)}(\boldsymbol{u}^*) - X^*$, $D\mathcal{A}_K(\boldsymbol{x}^{(K)}(\boldsymbol{u}^*), \boldsymbol{u}^*) - D\mathcal{A}_K(\boldsymbol{x}^*, \boldsymbol{u}^*)$, and the derivative of the solution $X_*$. All of these terms are computed at $k = K$ and therefore, are fixed. The explicit expressions can be found in the proof.

Owing to Corollary A.7 and our discussion in Remark 2.1, the following result shows that the work of Beck (1994) is reproduced and strengthened with a convergence rate guarantee under the additional requirement of Assumption 2.1(ii).

**Corollary 2.9.** *The conclusions of Theorems 2.5 and 2.7 also hold when, in their respective hypotheses, Assumption 2.1(iii) is replaced by "there exists a $C^1$-smooth $\mathcal{A} \colon V \to \mathcal{X}$, such that $D\mathcal{A}_k(\boldsymbol{x}^*, \boldsymbol{u}^*)$ converges to $D\mathcal{A}(\boldsymbol{x}^*, \boldsymbol{u}^*)$ and $\rho(D_{\boldsymbol{x}}\mathcal{A}(\boldsymbol{x}^*, \boldsymbol{u}^*)) < 1$".*

*Remark* 2.10. The conclusions of Theorems 2.5 and 2.7 and Corollary 2.9 do not require the neighbourhood $V$ of $(\boldsymbol{x}^*, \boldsymbol{u}^*)$ to be a connected set as long as Assumptions 2.2 and 2.3 are respectively satisfied along with Assumption 2.1(iv).

## 3. Differentiation of Proximal Gradient-type Methods

We now turn our attention to unrolling PGD with variable step size and APG when solving

$$\min_{\boldsymbol{x} \in \mathcal{X}} F(\boldsymbol{x}, \boldsymbol{u}), \quad F := f + g, \tag{8}$$

and aim at providing convergence (rate) guarantees for the corresponding derivative sequences. The idea is to consider the class of partly smooth functions (Lewis, 2002) which models a wide range of non-smooth regularizers and constraints appearing in practice (Vaiter et al., 2017). Furthermore, this setting (see Assumption 3.1) under some additional assumptions (see Assumptions 3.2 and 3.3) yields a more general IFT (see Theorem 3.1) for differentiating the solution mapping of (8). This in turn allows us to differentiate the update mappings of PGD (see Theorem 3.4) and APG (see Theorem 3.10), since they are also the solution mappings of partly smooth functions (see Equations 3, 10 and 13).

Examples of partly smooth functions include norms defined on vector space including $\ell_1$, $\ell_2$, and $\ell_\infty$ norms, as well as their combinations. These norms are commonly used as regularizers in machine learning: $\ell_1$ and $\ell_\infty$ promote sparsity and anti-sparsity, respectively, while the combined $\ell_{2,1}$ norm induces group sparsity. Analogous norms defined on

matrix space which include nuclear norm, Frobenius norm, and spectral norm and their combinations are also partly smooth and promote similar forms of strcutred-sparsity for matrices. The formal definition of partial smoothness is provided in Section A.5 in the appendix.

## 3.1. Problem Setting

We first define our setting in a more precise manner. Let $\mathcal{M}$ be a $C^2$-smooth manifold and $\Omega \subset \mathcal{U}$ be an open set, we need $f$ and $g$ to satisfy the following assumption.

**Assumption 3.1** (Convex Partly Smooth Objective). $f\colon \mathcal{X} \times \mathcal{U} \to \mathbb{R}$ is $C^2$-smooth, $g\colon \mathcal{X} \times \mathcal{U} \to \overline{\mathbb{R}}$ is partly smooth relative to $\mathcal{M} \times \Omega$ and for every $\boldsymbol{u} \in \Omega$, $f(\cdot, \boldsymbol{u})$ and $g(\cdot, \boldsymbol{u})$ are convex, $f(\cdot, \boldsymbol{u})$ has an $L$-Lipschitz continuous gradient, and $g(\cdot, \boldsymbol{u})$ has a simple proximal mapping.

When Assumption 3.1 is satisfied, we use Mehmood & Ochs (2022, Remark 22(iv)) to denote the Riemannian gradient and Hessian of $F$. Let $\boldsymbol{x}^* \in \mathcal{M}$ be a solution of (8) at $\boldsymbol{u}^* \in \Omega$. When $g = 0$, IFT requires $\nabla_{\boldsymbol{x}}^2 f(\boldsymbol{x}^*, \boldsymbol{u}^*) \succ 0$ which is also a sufficient condition for the linear convergence of gradient descent and the Heavy-ball method (Polyak, 1987). When $g$ is partly smooth, IFT requires positive definiteness of $\nabla_{\mathcal{M}}^2 F(\boldsymbol{x}^*, \boldsymbol{u}^*)$ (Lewis, 2002) while the linear convergence of PGD and APG is established by the positive definiteness of $\Pi(\boldsymbol{x}^*)\nabla_{\boldsymbol{x}}^2 f(\boldsymbol{x}^*, \boldsymbol{u}^*)\Pi(\boldsymbol{x}^*)$ (Liang et al., 2014; 2017).

**Assumption 3.2** (Restricted Positive Definiteness). Let $(\boldsymbol{x}^*, \boldsymbol{u}^*) \in \mathcal{M} \times \Omega$ be a given point.

(i) The Hessian $\nabla_{\mathcal{M}}^2 F(\boldsymbol{x}^*, \boldsymbol{u}^*)$ is positive definite on $T_{\boldsymbol{x}^*}\mathcal{M}$, that is,

$$\nabla_{\mathcal{M}}^2 F(\boldsymbol{x}^*, \boldsymbol{u}^*) \succ 0. \tag{RPD-i}$$

(ii) Moreover, $\Pi(\boldsymbol{x}^*)\nabla_{\boldsymbol{x}}^2 f(\boldsymbol{x}^*, \boldsymbol{u}^*)$ is positive definite on $T_{\boldsymbol{x}^*}\mathcal{M}$, that is,

$$\Pi(\boldsymbol{x}^*)\nabla_{\boldsymbol{x}}^2 f(\boldsymbol{x}^*, \boldsymbol{u}^*)\big|_{T_{\boldsymbol{x}^*}\mathcal{M}} \succ 0. \tag{RPD-ii}$$

Finally, for partly smooth problems, IFT and linear convergence of PGD and APG also require a non-degeneracy assumption to hold which ensures that the solution stays in $\mathcal{M}$ for any $\boldsymbol{u}$ near $\boldsymbol{u}^*$.

**Assumption 3.3** (Non-degeneracy). The non-degeneracy condition is satisfied at $(\boldsymbol{x}^*, \boldsymbol{u}^*) \in \mathcal{M} \times \Omega$, that is,

$$0 \in \operatorname{ri} \partial_{\boldsymbol{x}} F(\boldsymbol{x}^*, \boldsymbol{u}^*). \tag{ND}$$

In (ND), $\operatorname{ri}$ denotes the relative interior of a convex set.

## 3.2. Implicit Function Theorem

The following theorem is thanks to Lewis (2002) which furnishes a $C^1$-smooth solution map for (8) near $\boldsymbol{u}^*$. The expression for the derivative can be found in Vaiter et al. (2017); Mehmood & Ochs (2022).

**Theorem 3.1** (Differentiation of Solution Map). *Let $f$ and $g$ satisfy Assumption 3.1 and $(\boldsymbol{x}^*, \boldsymbol{u}^*) \in \mathcal{M} \times \Omega$ be such that Assumptions 3.2(i) and 3.3 are satisfied. Then there exist an open neighbourhood $U \subset \Omega$ of $\boldsymbol{u}^*$ and a continuously differentiable mapping $\psi\colon U \to \mathcal{M}$ such that for all $\boldsymbol{u} \in U$,*

*(i) $\psi(\boldsymbol{u})$ is the unique minimizer of $F(\cdot, \boldsymbol{u})|_{\mathcal{M}}$,*

*(ii) (ND) and (RPD-i) are satisfied at $(\psi(\boldsymbol{u}), \boldsymbol{u})$, and*

*(iii) the derivative of $\psi$ is given by*

$$D\psi(\boldsymbol{u}) = -\nabla_{\mathcal{M}}^2 F(\psi(\boldsymbol{u}), \boldsymbol{u})^\dagger D_{\boldsymbol{u}} \nabla_{\mathcal{M}} F(\psi(\boldsymbol{u}), \boldsymbol{u}), \tag{9}$$

*where $\nabla_{\mathcal{M}}^2 F(\psi(\boldsymbol{u}), \boldsymbol{u})^\dagger$ denotes the pseudoinverse of $\nabla_{\mathcal{M}}^2 F(\psi(\boldsymbol{u}), \boldsymbol{u})$.*

## 3.3. The Extrapolated Proximal Gradient Algorithm

Although Theorem 3.1 is not practical when computing the derivative of the solution map, it allows us to differentiate the update mappings of PGD and APG as we will see shortly. Algorithm 1 provides the update procedure for proximal gradient with extrapolation where $\mathrm{P}_{\alpha g}$ is defined in (3). This algorithm reduces to (i) PGD when $\beta_k = 0$ for all $k \in \mathbb{N}$, and (ii) APG when the sequence $\beta_k$ ensures acceleration in PGD (Beck & Teboulle, 2009; Chambolle & Dossal, 2015).

---

**Algorithm 1** Proximal Gradient with Extrapolation

- **Initialization:** $\boldsymbol{x}^{(0)} = \boldsymbol{x}^{(-1)} \in \mathcal{X}$, $\boldsymbol{u} \in \mathcal{U}$, $0 < \underline{\alpha} \leq \bar{\alpha} < 2/L$.

- **Parameter:** $(\alpha_k)_{k\in\mathbb{N}} \in [\underline{\alpha}, \bar{\alpha}]$ and $(\beta_k)_{k\in\mathbb{N}} \in [0, 1]$.

- **Update $k \geq 0$:**

$$\begin{aligned} \boldsymbol{y}^{(k)} &:= (1 + \beta_k)\boldsymbol{x}^{(k)} - \beta_k \boldsymbol{x}^{(k-1)} \\ \boldsymbol{w}^{(k)} &:= \boldsymbol{y}^{(k)} - \alpha_k \nabla_{\boldsymbol{x}} f(\boldsymbol{y}^{(k)}, \boldsymbol{u}) \\ \boldsymbol{x}^{(k+1)} &:= \mathrm{P}_{\alpha_k g}(\boldsymbol{w}^{(k)}, \boldsymbol{u}). \end{aligned} \tag{APG}$$

---

## 3.4. Proximal Gradient Descent

We first extend our results from Section 2 and provide convergence rate guarantees for AD of PGD. For a given step size $\alpha > 0$, we write the update mapping for PGD in a more

compact manner through the map $\mathcal{A}_\alpha : \mathcal{X} \times \Omega \to \mathcal{X}$ defined by

$$\mathcal{A}_\alpha(\boldsymbol{x}, \boldsymbol{u}) := \mathrm{P}_{\alpha g}(\boldsymbol{x} - \alpha \nabla_{\boldsymbol{x}} f(\boldsymbol{x}, \boldsymbol{u}), \boldsymbol{u}), \qquad (10)$$

Liang et al. (2014) established that under Assumptions 3.1, 3.2(ii) and 3.3, all the iterates $\boldsymbol{x}^{(k)}(\boldsymbol{u}^*)$ of PGD lie on $\mathcal{M}$ after a finite number of iterations and exhibit a local linear convergence behaviour as summarized by the following lemma.

**Lemma 3.2** (Activity Identification and Linear Convergence of PGD)**.** *Let $f$ and $g$ satisfy Assumption 3.1 and $(\boldsymbol{x}^*, \boldsymbol{u}^*) \in \mathcal{M} \times \Omega$ be such that Assumption 3.3 is satisfied. For $\alpha_k \in [\underline{\alpha}, \bar{\alpha}]$ and $\beta_k := 0$, let the sequence $(\boldsymbol{x}^{(k)}(\boldsymbol{u}^*))_{k \in \mathbb{N}}$ generated by Algorithm 1 converges to $\boldsymbol{x}^*$. Then there exists $K \in \mathbb{N}$, such that $\boldsymbol{x}^{(k)}(\boldsymbol{u}^*) \in \mathcal{M}$ for all $k \geq K$. Moreover when Assumption 3.2(ii) is also satisfied and $\bar{\alpha} < 2M/L^2$ where $M := \lambda_{\max}(\Pi(\boldsymbol{x}^*)\nabla_{\boldsymbol{x}}^2 f(\boldsymbol{x}^*, \boldsymbol{u}^*)\Pi(\boldsymbol{x}^*))$, then $\boldsymbol{x}^{(k)}(\boldsymbol{u}^*)$ converge linearly to $\boldsymbol{x}^*$.*

*Remark* 3.3. The precise rate of convergence of PGD can be found in Liang et al. (2014, Theorem 3.1).

Mehmood & Ochs (2022, Theorem 34) provide a foundation for proving the following crucial differentiablity result for $\mathcal{A}_\alpha$ by combining Theorem 3.1 and Lemma 3.2. However, the result below is novel since it establishes differentiability of $\mathcal{A}_{\alpha_k}$ at the iterates $\boldsymbol{x}^{(k)}(\boldsymbol{u}^*)$ of PGD for all $k \in \mathbb{N}$ and also shows that the derivative sequence $D\mathcal{A}_{\alpha_k}$ is well-behaved.

**Theorem 3.4.** *Let $f$ and $g$ satisfy Assumption 3.1 and $(\boldsymbol{x}^*, \boldsymbol{u}^*) \in \mathcal{M} \times \Omega$ be such that Assumption 3.3 are satisfied. For $\alpha_k \in [\underline{\alpha}, \bar{\alpha}]$ and $\beta_k := 0$, let the sequence $(\boldsymbol{x}^{(k)} := \boldsymbol{x}^{(k)}(\boldsymbol{u}^*))_{k \in \mathbb{N}}$ generated by Algorithm 1 converges to $\boldsymbol{x}^*$. Then there exists $K \in \mathbb{N}$, such that,*

(i) *the mapping $(\boldsymbol{x}, \boldsymbol{u}, \alpha) \mapsto \mathcal{A}_\alpha(\boldsymbol{x}, \boldsymbol{u})$ defined in (10) is $C^1$-smooth near $(\boldsymbol{x}^{(k)}, \boldsymbol{u}^*, \alpha_k)$ and $(\boldsymbol{x}^*, \boldsymbol{u}^*, \alpha_k)$ for all $k \geq K$,*

(ii) *$(D\mathcal{A}_{\alpha_k}(\boldsymbol{x}^{(k)}, \boldsymbol{u}^*) - D\mathcal{A}_{\alpha_k}(\boldsymbol{x}^*, \boldsymbol{u}^*))_{k \geq K}$ converges to $0$, and*

(iii) *additionally, when $\mathcal{M}$ is $C^3$-smooth, $\nabla_{\mathcal{M}} g$, $\nabla_{\boldsymbol{x}}^2 f$, $\nabla_{\mathcal{M}}^2 g$, $D_{\boldsymbol{u}} \nabla_{\boldsymbol{x}} f$, and $D_{\boldsymbol{u}} \nabla_{\mathcal{M}} g$ are locally Lipschitz continuous near $(\boldsymbol{x}^*, \boldsymbol{u}^*)$ and $(\boldsymbol{x}^{(k)})_{k \in \mathbb{N}}$ converges linearly, then $D\mathcal{A}_{\alpha_k}(\boldsymbol{x}^{(k)}, \boldsymbol{u}^*) - D\mathcal{A}_{\alpha_k}(\boldsymbol{x}^*, \boldsymbol{u}^*) = \mathcal{O}(\boldsymbol{x}^{(k)} - \boldsymbol{x}^*)$.*

*Proof.* The proof is in Section C.1 in the appendix. $\square$

*Remark* 3.5. The $C^3$-smoothness of $\mathcal{M}$ is a sufficient condition for the local Lipschitz continuity of the Weingarten map $(\boldsymbol{x}, \boldsymbol{v}) \mapsto \mathfrak{W}_{\boldsymbol{x}}(\cdot, \boldsymbol{v})$.

### 3.4.1. IMPLICIT DIFFERENTIATION

Under Assumptions 3.1–3.3, Mehmood & Ochs (2022, Theorem 34) provided an IFT for the fixed-point equation of PGD which is more practical for using ID than (9) and is restated below.

**Theorem 3.6.** *Let $f$ and $g$ satisfy Assumption 3.1 and $(\boldsymbol{x}^*, \boldsymbol{u}^*) \in \mathcal{M} \times \Omega$ be such that Assumptions 3.2(ii) and 3.3 are satisfied. Then for any $\alpha \in [\underline{\alpha}, \bar{\alpha}]$, $\rho(D_{\boldsymbol{x}}\mathcal{A}_\alpha(\boldsymbol{x}^*, \boldsymbol{u}^*)\Pi(\boldsymbol{x}^*)) < 1$. Additionally when Assumption 3.2(i) is also satisfied, the (possibly reduced) neighbourhood $U$ and the mapping $\psi$ from Theorem 3.1 satisfy $\boldsymbol{x} = \mathcal{A}_\alpha(\boldsymbol{x}, \boldsymbol{u})$ and*

$$D\psi(\boldsymbol{u}) = (I - D_{\boldsymbol{x}}\mathcal{A}_\alpha(\boldsymbol{x}, \boldsymbol{u})\Pi(\boldsymbol{x}))^{-1} D_{\boldsymbol{u}}\mathcal{A}_\alpha(\boldsymbol{x}, \boldsymbol{u}),$$
(11)

*for all $\boldsymbol{u} \in U$ and $\boldsymbol{x} := \psi(\boldsymbol{u})$.*

### 3.4.2. AUTOMATIC DIFFERENTIATION

Using Theorems 3.4, and 3.6 and our results from Section 2, we obtain the convergence (rate) guarantees for AD of PGD.

**Theorem 3.7.** *Let $f$ and $g$ satisfy Assumption 3.1 and $(\boldsymbol{x}^*, \boldsymbol{u}^*) \in \mathcal{M} \times \Omega$ be such that Assumptions 3.2 and 3.3 are satisfied. Let $\alpha_k \in [\underline{\alpha}, \bar{\alpha}]$, $\beta_k := 0$, the sequence $(\boldsymbol{x}^{(k)}(\boldsymbol{u}^*))_{k \in \mathbb{N}}$ generated by Algorithm 1 converges to $\boldsymbol{x}^*$ with $\boldsymbol{x}^{(0)}$ sufficiently close to $\boldsymbol{x}^*$, and $\limsup_{k \to \infty} \rho(D_{\boldsymbol{x}}\mathcal{A}_{\alpha_k}(\boldsymbol{x}^*, \boldsymbol{u}^*)) < 1$. Then the sequence $(D\boldsymbol{x}^{(k)}(\boldsymbol{u}^*))_{k \in \mathbb{N}}$ converges to $D\psi(\boldsymbol{u}^*)$. Additionally $(D\boldsymbol{x}^{(k)}(\boldsymbol{u}^*))_{k \in \mathbb{N}}$ converges linearly with rate $\max(\rho + \delta, q_{\boldsymbol{x}})$ where $\rho := \limsup_{k \to \infty} \rho(D_{\boldsymbol{x}}\mathcal{A}_{\alpha_k}(\boldsymbol{x}^*, \boldsymbol{u}^*))$, when $\boldsymbol{x}^{(k)}(\boldsymbol{u}^*)$ converges linearly with rate $q_{\boldsymbol{x}} < 1$, $\mathcal{M}$ is $C^3$-smooth, and $\nabla_{\mathcal{M}} g$, $\nabla_{\boldsymbol{x}}^2 f$, $\nabla_{\mathcal{M}}^2 g$, $D_{\boldsymbol{u}} \nabla_{\boldsymbol{x}} f$, and $D_{\boldsymbol{u}} \nabla_{\mathcal{M}} g$ are locally Lipschitz continuous near $(\boldsymbol{x}^*, \boldsymbol{u}^*)$. In particular, for all $\delta \in (0, 1 - \rho)$, there exist $\xi_1$, $\xi_2$, and $K \in \mathbb{N}$, such that for all $k \geq K$, we have*

$$\|D\boldsymbol{x}^{(k)}(\boldsymbol{u}^*) - D\psi(\boldsymbol{u}^*)\| \leq \xi_1(\rho + \delta)^{k-K}$$
$$+ \xi_2(k - K)\max(\rho + \delta, q_{\boldsymbol{x}})^{k-K}. \qquad (12)$$

*Proof.* The proof is in Section C.2 in the appendix. $\square$

*Remark* 3.8. (i) The constants $\xi_1$ and $\xi_2$ in (12) depend on derivative errors $D\boldsymbol{x}^{(K)}(\boldsymbol{u}^*) - D\psi(\boldsymbol{u}^*)$, $D\mathcal{A}_{\alpha_K}(\boldsymbol{x}^{(K)}(\boldsymbol{u}^*), \boldsymbol{u}^*) - D\mathcal{A}_{\alpha_K}(\boldsymbol{x}^*, \boldsymbol{u}^*)$, and the derivative of the solution $D\psi(\boldsymbol{u}^*)$ in the same manner as described in Remark 2.8.

(ii) Because $\boldsymbol{x}^{(0)}$ is not close enough to $\boldsymbol{x}^*$ in practice, one may resort to late-starting (Mehmood & Ochs, 2022, Section 1.4.2).

(iii) In practice, we do not need $\boldsymbol{x}^{(0)}$ to be close enough to $\boldsymbol{x}^*$ because even when the update map $\mathcal{A}_{\alpha_k}$ is not differentiable in the earlier iterations of Algorithm 1,

the autograd libraries still yield a finite output as the derivative (Bolte & Pauwels, 2020; 2021). Hence, AD of Algorithm 1 can still recover a good estimate of $D\psi(\boldsymbol{u})$ as long as Algorithm 1 is run for sufficiently large number of iterations (see also Mehmood & Ochs, 2022, Remark 45(i)).

(iv) When $\alpha_k$ is generated through line-search methods, it also depends on $\boldsymbol{u}$ in a possibly non-differentiable way. Therefore, the total derivative of $\boldsymbol{x}^{(k)}$ with respect to $\boldsymbol{u}$ may not make sense. However computing $D\boldsymbol{x}^{(k)}(\boldsymbol{u})$ by ignoring this dependence — for instance, in practice, through routines like stop_gradient (Bradbury et al., 2018; Abadi et al., 2016) and detach (Paszke et al., 2019) — the true derivative is still recovered in the limit provided that conditions of Theorem 3.7 are met. This fact was first explored in Griewank et al. (1993).

(v) All of the non-smooth functions provided in, for instance, Vaiter et al. (2017) satisfy the local Lipschitz continuity condition of $\nabla_{\mathcal{M}}g$, and $\nabla^2_{\mathcal{M}}g$ and the $C^3$ smoothness of $\mathcal{M}$.

(vi) The conditions like the linear convergence of $\boldsymbol{x}^{(k)}(\boldsymbol{u}^*)$ and $\limsup_{k\to\infty} \rho(D_{\boldsymbol{x}}\mathcal{A}_{\alpha_k}(\boldsymbol{x}^*,\boldsymbol{u}^*)) < 1$ are not required if the step size sequence $\alpha_k$ is bounded as in the hypothesis of Lemma 3.2.

### 3.5. Accelerated Proximal Gradient

We similarly compute AD of APG and show (linear) convergence of the corresponding derivative iterates. Given step size $\alpha > 0$ and extrapolation parameter $\beta \in [0,1]$, we define the update mapping $\mathcal{A}_{\alpha,\beta}\colon \mathcal{X} \times \mathcal{X} \times \mathcal{U} \to \mathcal{X} \times \mathcal{X}$ by

$$\mathcal{A}_{\alpha,\beta}(\boldsymbol{z},\boldsymbol{u}) := \left(\mathcal{A}_{\alpha}\left(\boldsymbol{x}_1 + \beta(\boldsymbol{x}_1 - \boldsymbol{x}_2), \boldsymbol{u}\right), \boldsymbol{x}_1\right), \quad (13)$$

for $\boldsymbol{z} := (\boldsymbol{x}_1, \boldsymbol{x}_2) \in \mathcal{X} \times \mathcal{X}$. Just like PGD, APG also exhibit activity identification property and local linear convergence under Assumptions 3.1, 3.2(ii), and 3.3 (Liang et al., 2017).

**Lemma 3.9** (Activity Identification and Linear Convergence of APG). *Let $f$ and $g$ satisfy Assumption 3.1 and $(\boldsymbol{x}^*, \boldsymbol{u}^*) \in \mathcal{M} \times \Omega$ be such that Assumption 3.3 is satisfied. For $\alpha_k \in [\underline{\alpha}, \bar{\alpha}]$ and $\beta_k \in [0,1]$, let the sequence $(\boldsymbol{x}^{(k)}(\boldsymbol{u}^*))_{k\in\mathbb{N}}$ generated by Algorithm 1 converges to $\boldsymbol{x}^*$. Then there exists $K \in \mathbb{N}$, such that $\boldsymbol{x}^{(k)}(\boldsymbol{u}^*) \in \mathcal{M}$ for all $k \geq K$. Moreover when Assumption 3.2(ii) is also satisfied, $\alpha_k \to \alpha_*$ and $\beta_k \to \beta_*$ such that $-1/(1 + 2\beta_*) < \lambda_{\min}(D_{\boldsymbol{x}}\mathcal{A}_{\alpha_*}(\boldsymbol{x}^*, \boldsymbol{u}^*))$, then $\boldsymbol{x}^{(k)}(\boldsymbol{u}^*)$ converge linearly to $\boldsymbol{x}^*$ with rate $\rho(D_{\boldsymbol{z}}\mathcal{A}_{\alpha_*,\beta_*}(\boldsymbol{x}^*, \boldsymbol{x}^*, \boldsymbol{u}^*)\Pi(\boldsymbol{x}^*, \boldsymbol{x}^*))$.*

Using Theorem 3.1, and Lemma 3.9, we can differentiate $\mathcal{A}_{\alpha_k,\beta_k}$ near $(\boldsymbol{x}^*, \boldsymbol{x}^*, \boldsymbol{u}^*)$ for all $k \in \mathbb{N}$. The following

result is mostly derived from Mehmood & Ochs (2022, Theorem 39).

**Theorem 3.10.** *Let $f$ and $g$ satisfy Assumption 3.1 and $(\boldsymbol{x}^*, \boldsymbol{u}^*) \in \mathcal{M} \times \Omega$ be such that Assumption 3.3 are satisfied. For $[\underline{\alpha}, \bar{\alpha}] \ni \alpha_k \to \alpha_*$ and $[0,1] \ni \beta_k \to \beta_*$, let the sequence $(\boldsymbol{x}^{(k)} := \boldsymbol{x}^{(k)}(\boldsymbol{u}^*))_{k\in\mathbb{N}}$ generated by Algorithm 1 converges to $\boldsymbol{x}^*$. Then there exist $V_{\alpha_*} \in \mathcal{N}_{(\boldsymbol{x}^*,\boldsymbol{x}^*,\boldsymbol{u}^*,\alpha_*)}$ and $K \in \mathbb{N}$, such that*

(i) *the mapping $(\boldsymbol{z}, \boldsymbol{u}, \alpha) \mapsto \mathcal{A}_{\alpha,\beta}(\boldsymbol{z}, \boldsymbol{u})$ defined in (13) is $C^1$-smooth on $V_{\alpha_*}$ and $(\boldsymbol{z}^{(k)}, \boldsymbol{u}^*, \alpha_k) \in V_{\alpha_*}$ for all $k \geq K$,*

(ii) $(D\mathcal{A}_{\alpha_k,\beta_k}(\boldsymbol{z}^{(k)}, \boldsymbol{u}^*) - D\mathcal{A}_{\alpha_k,\beta_k}(\boldsymbol{z}^*, \boldsymbol{u}^*))_{k\geq K}$ *converges to $0$, and $(D\mathcal{A}_{\alpha_k,\beta_k}(\boldsymbol{z}^*, \boldsymbol{u}^*))_{k\geq K}$ converges to $D\mathcal{A}_{\alpha_*,\beta_*}(\boldsymbol{z}^*, \boldsymbol{u}^*)$, and*

(iii) *additionally, when $\mathcal{M}$ is $C^3$-smooth, $\nabla_{\mathcal{M}}g$, $\nabla^2_{\boldsymbol{x}}f$, $\nabla^2_{\mathcal{M}}g$, $D_{\boldsymbol{u}}\nabla_{\boldsymbol{x}}f$, and $D_{\boldsymbol{u}}\nabla_{\mathcal{M}}g$ are locally Lipschitz continuous near $(\boldsymbol{x}^*, \boldsymbol{u}^*)$ and $(\boldsymbol{x}^{(k)})_{k\in\mathbb{N}}$ converges linearly, then $D\mathcal{A}_{\alpha_k,\beta_k}(\boldsymbol{z}^{(k)}, \boldsymbol{u}^*) - D\mathcal{A}_{\alpha_k,\beta_k}(\boldsymbol{z}^*, \boldsymbol{u}^*) = \mathcal{O}(\boldsymbol{x}^{(k)} - \boldsymbol{x}^*)$,*

*where $\boldsymbol{z} := (\boldsymbol{x}_1, \boldsymbol{x}_2)$, $\boldsymbol{z}^{(k)} := (\boldsymbol{x}^{(k)}, \boldsymbol{x}^{(k-1)})$, and $\boldsymbol{z}^* := (\boldsymbol{x}^*, \boldsymbol{x}^*)$*

*Proof.* The proof is in Section C.3 in the appendix. $\square$

#### 3.5.1. IMPLICIT DIFFERENTIATION

Similarly Theorems 3.1, and 3.10 can be used to yield an IFT for the fixed-point equation of APG (Mehmood & Ochs, 2022, Theorem 39), which we recall below.

**Theorem 3.11.** *Let $f$ and $g$ satisfy Assumption 3.1 and $(\boldsymbol{x}^*, \boldsymbol{u}^*) \in \mathcal{M} \times \Omega$ be such that Assumptions 3.2(ii) and 3.3 are satisfied. Then for any $\alpha \in [\underline{\alpha}, \bar{\alpha}]$ and $\beta \in [0,1]$ with $-1/(1 + 2\beta) < \lambda_{\min}(D_{\boldsymbol{x}}\mathcal{A}_{\alpha}(\boldsymbol{x}^*, \boldsymbol{u}))$, we have $\rho(D_{\boldsymbol{z}}\mathcal{A}_{\alpha,\beta}(\boldsymbol{x}^*, \boldsymbol{x}^*, \boldsymbol{u}^*)\Pi(\boldsymbol{x}^*, \boldsymbol{x}^*)) < 1$. Additionally when Assumption 3.2(i) is also satisfied, the (possibly reduced) neighbourhood $U$ and the mapping $\psi$ from Theorem 3.1 satisfy $\boldsymbol{z} = \mathcal{A}_{\alpha,\beta}(\boldsymbol{z}, \boldsymbol{u})$ and*

$$\begin{bmatrix} D\psi(\boldsymbol{u}) \\ D\psi(\boldsymbol{u}) \end{bmatrix} = \left(I - D_{\boldsymbol{z}}\mathcal{A}_{\alpha,\beta}(\boldsymbol{z}, \boldsymbol{u})\Pi(\boldsymbol{z})\right)^{-1} D_{\boldsymbol{u}}\mathcal{A}_{\alpha,\beta}(\boldsymbol{z}, \boldsymbol{u}),$$
$$(14)$$

*for all $\boldsymbol{u} \in U$ and $\boldsymbol{z} := (\psi(\boldsymbol{u}), \psi(\boldsymbol{u}))$.*

#### 3.5.2. AUTOMATIC DIFFERENITATION

Mehmood & Ochs (2022, see Theorem 44) established the convergence of the derivative iterates of APG. We strengthen their results by providing convergence rate guarantees in our final result below.

**Theorem 3.12.** *Let $f$ and $g$ satisfy Assumption 3.1 and $(\boldsymbol{x}^*, \boldsymbol{u}^*) \in \mathcal{M} \times \Omega$ be such that Assumption 3.3 are satisfied. For $[\underline{\alpha}, \bar{\alpha}] \ni \alpha_k \to \alpha_*$ and $[0, 1] \ni \beta_k \to \beta_*$ such that $-1/(1 + 2\beta_*) < \lambda_{\min}(D_{\boldsymbol{x}} \mathcal{A}_{\alpha_*}(\boldsymbol{x}^*, \boldsymbol{u}^*))$, let the sequence $(\boldsymbol{x}^{(k)}(\boldsymbol{u}^*))_{k \in \mathbb{N}}$ generated by Algorithm 1 converges to $\boldsymbol{x}^*$ with $\boldsymbol{x}^{(0)}$ sufficiently close to $\boldsymbol{x}^*$. Then the sequence $(D\boldsymbol{x}^{(k)}(\boldsymbol{u}^*))_{k \in \mathbb{N}}$ converges to $D\psi(\boldsymbol{u}^*)$. Additionally, $(D\boldsymbol{x}^{(k)}(\boldsymbol{u}^*))_{k \in \mathbb{N}}$ converges linearly with rate $\max(\rho + \delta, q_{\boldsymbol{x}})$ where $\rho \coloneqq \rho(D_{\boldsymbol{z}} \mathcal{A}_{\alpha_*, \beta_*}(\boldsymbol{x}^*, \boldsymbol{u}^*))$, when $\boldsymbol{x}^{(k)}(\boldsymbol{u}^*)$ converges linearly with rate $q_{\boldsymbol{x}} < 1$, $\mathcal{M}$ is $C^3$-smooth, and $\nabla_{\mathcal{M}} g$, $\nabla_{\boldsymbol{x}}^2 f$, $\nabla_{\mathcal{M}}^2 g$, $D_{\boldsymbol{u}} \nabla_{\boldsymbol{x}} f$, and $D_{\boldsymbol{u}} \nabla_{\mathcal{M}} g$ are locally Lipschitz continuous near $(\boldsymbol{x}^*, \boldsymbol{u}^*)$. In particular, for all $\delta \in (0, 1 - \rho)$, there exist $\xi_1, \xi_2,$ and $K \in \mathbb{N}$, such that for all $k \geq K$, we have*

$$
\begin{aligned}
\|D\boldsymbol{x}^{(k)}(\boldsymbol{u}^*) - D\psi(\boldsymbol{u}^*)\| &\leq \xi_1 (\rho + \delta)^{k-K} \\
&+ \xi_2 (k - K) \max(\rho + \delta, q_{\boldsymbol{x}})^{k-K}.
\end{aligned} \tag{15}
$$

*Proof.* The proof is in Section C.4 in the appendix. □

*Remark* 3.13. The arguments made in Remark 3.8 naturally extend to Theorem 3.12.

# 4. Experiments

To test our results, we provide numerical demonstration on a few examples from classical Machine Learning. These include lasso regression, that is,

$$
\min_{\boldsymbol{w} \in \mathbb{R}^N} \frac{1}{2} \|X\boldsymbol{w} - \boldsymbol{y}\|_2^2 + \lambda \|\boldsymbol{w}\|_1 , \tag{16}
$$

with parameters $\boldsymbol{u} \coloneqq (X, \boldsymbol{y}, \lambda)$, binary classification using $\ell_2$-regularized logistic regression, that is,

$$
\min_{\boldsymbol{w} \in \mathbb{R}^N} \frac{1}{M} \sum_{i=1}^{M} \log\left(1 + \exp(-y_i \mathbf{x}_i^T \boldsymbol{w})\right) + \frac{\lambda}{2} \|\boldsymbol{w}\|_2^2, \tag{17}
$$

with parameters $\boldsymbol{u} \coloneqq (X, \lambda)$, and finally multiclass classification using $\ell_2$-regularized multinomial logistic regression, that is,

$$
\min_{W \in \mathbb{R}^{N \times C}} -\frac{1}{M} \sum_{i=1}^{M} \log(p_{y_i}(\boldsymbol{x}_i)) + \frac{\lambda}{2} \|W\|_F^2 , \tag{18}
$$

with parameters $\boldsymbol{u} \coloneqq (X, \lambda)$. In (18), $\|\cdot\|_F$ is the Frobenius norm and $p_j(\boldsymbol{x})$ denotes the $j^{\text{th}}$ element of the predicted output $\boldsymbol{p}(\boldsymbol{x}) \coloneqq \sigma(W^T \boldsymbol{x})$ for each input $\boldsymbol{x} \in \mathbb{R}^N$ with $\sigma$ being the softmax activation function. In all the problems, $X \in \mathbb{R}^{M \times N}$ and $\boldsymbol{y} \in \mathbb{R}^M$ represent the dataset and the parameters are selected in such a way that Assumptions 3.1–3.3 are satisfied.

We solve the three problems through PGD with four different choices of step sizes and APG with fixed step size and

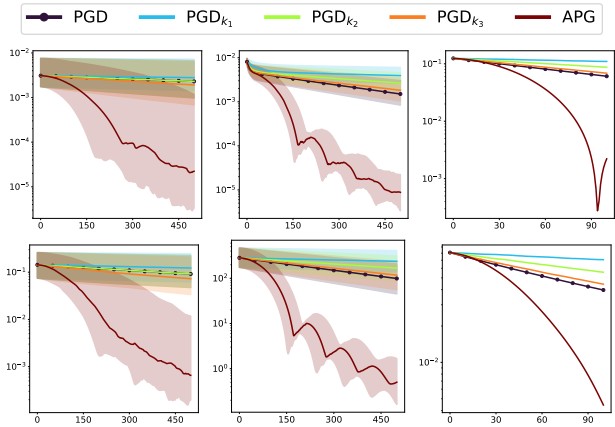

*Figure 1.* Error plots of iterates (top row) of PGD and APG for lasso (left column), logistic (middle column), and multinomial logistic (right column) regression along with their derivative iterates (bottom row). Similarity in the convergence rates of the original and the derivative iterates is clearly visible.

$\beta_k \coloneqq (k-1)/(k+5)$ (depicted by APG in Figure 1). This generates five different algorithm sequences $(\boldsymbol{x}^{(k)}(\boldsymbol{u}))_{k \in \mathbb{N}}$ for each problem. In Figure 1, PGD represents step size $\alpha_k$ set to fixed optimal value $\alpha_*$, whereas $\text{PGD}_{k_i}$ represents varying step size $\alpha_k$ drawn from uniform distribution with parameters $\frac{2(i-1)}{3L}$, and $\frac{2i}{3L}$ for all $k \in \mathbb{N}$ and for all $i \in \{1, 2, 3\}$. We solve (16) for 50 randomly generated datasets, (17) for 50 perturbed instances of MADELON dataset (Dua & Graff, 2017), and (18) for a single instance of CIFAR10 dataset (Krizhevsky, 2009). For each problem, we initialize $\boldsymbol{w}^{(0)} \in B_{10^{-2}}(\boldsymbol{w}^*)$ by first solving it partially.

In Figure 1, the top row shows the median error plots of the five algorithms and the bottom row shows the errors of the corresponding derivatives with the same colour. The three columns correspond to the error plots for (16), (17), and (18) respectively. The figure clearly shows that the derivative error decays as fast as the algorithm error for each problem. In particular, the linear convergence behaviour of AD of APG for all the problems and $\text{PGD}_{k_i}$ with $i \in \{1, 2, 3\}$ for the lasso problem are only explained by our results.

# 5. Conclusion

We applied automatic differentiation on the iterative processes with time-varying update mappings. We strengthened a previous result (Beck, 1994) with convergence rate guarantee and extended them to a new setting. As an example for each setting, we adapted our results to proximal gradient descent with variable step size and its accelerated counterpart, that is, FISTA. We showed that the convergence rate of the algorithm is simply mirrored in its derivative iterates which was supported through experiments on toy problems.

## Acknowledgements

Sheheryar Mehmood and Peter Ochs are supported by the German Research Foundation (DFG Grant OC 150/4-1).

## Impact Statement

This paper presents work whose goal is to advance the field of Machine Learning. There are many potential societal consequences of our work, none which we feel must be specifically highlighted here.

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

# A. Preliminaries & Related Work

Before we move on to the proofs of our main results, we present some preliminary results which will be useful later. We also provide a recap of the results of Beck (1994) for a better understanding of our work.

## A.1. Real Analysis

In Section 2, we refer to equicontinuity and uniform Lipschitz continuity as cases where Assumptions 2.2 and 2.3 respectively hold. In this section, we provide defintion along with an example for each of these notions. Note that, in the definitions and examples below, $x^*$ is not necessarily the fixed-point of any of $\mathcal{A}_k(\cdot, u^*)$. Let $\mathcal{X}, \mathcal{U}$ be Euclidean spaces as in Section 1.4 and $\Omega \subset \mathcal{U}$ be an open set. We first define equicontinuity of a sequence of functions.

**Definition A.1** (Equicontinuity). A sequence of functions $(\mathcal{B}_k : \mathcal{X} \times \Omega \to \mathcal{X})_{k \in \mathbb{N}}$ with values $\mathcal{B}_k(x, u)$ is equicontinuous in $x$ at some $(x^*, u^*) \in \mathcal{X} \times \Omega$ if for any $\varepsilon > 0$, there exists a $\delta > 0$ such that $\|\mathcal{B}_k(x, u^*) - \mathcal{B}_k(x^*, u^*)\| < \varepsilon$ for all $k \in \mathbb{N}$ and for all $x \in \mathcal{X}$ such that $\|x - x^*\| < \delta$. The sequence $(\mathcal{B}_k)_{k \in \mathbb{N}}$ is equicontinuous in $x$ if it is equicontinuous in $x$ at every $(x^*, u^*) \in \mathcal{X} \times \Omega$.

Intuitively speaking, an equicontinuous sequence of functions behaves uniformly in terms of continuity because the neighbourhood $\{x : \|x - x^*\| < \delta\}$ in Definition A.1 does not depend on $k \in \mathbb{N}$. Below, we provide an example for equicontinuous functions.

**Example A.2.** *Let $F$ in ($\mathcal{P}$) be $C^2$-smooth. Given $(\alpha_k)_{k \in \mathbb{N}}$ such that $\alpha_k \leq \bar{\alpha}$ for all $k \in \mathbb{N}$ and some $\bar{\alpha} > 0$, we define $\mathcal{A}_k : \mathcal{X} \times \Omega \to \mathcal{X}$ by*

$$\mathcal{A}_k(x, u) := x - \alpha_k \nabla_x F(x, u), \tag{19}$$

*and $\mathcal{B}_k : \mathcal{X} \times \Omega \to \mathcal{X}$ by*

$$\mathcal{B}_k(x, u) := D\mathcal{A}_k(x, u) = \begin{bmatrix} I & 0 \end{bmatrix} - \alpha_k D(\nabla_x F)(x, u), \tag{20}$$

*where $D(\nabla_x F) \in \mathcal{L}(\mathcal{X} \times \mathcal{U}, \mathcal{X})$ is the derivative (with respect to both $x$ and $u$) of $\nabla_x F$ and $\begin{bmatrix} I & 0 \end{bmatrix} \in \mathcal{L}(\mathcal{X} \times \mathcal{U}, \mathcal{X})$ defines the canonical projection onto $\mathcal{X}$. Essentially, (19) defines $\mathcal{A}_k$ as the update mapping of gradient descent with bounded step size sequence while (20) defines $\mathcal{B}_k$ as its derivative with respect to $x$ and $u$. For any $x, x^* \in \mathcal{X}$ and $u^* \in \mathcal{U}$, we find that*

$$\mathcal{B}_k(x, u^*) - \mathcal{B}_k(x^*, u^*) = -\alpha_k \big( D(\nabla_x F)(x, u^*) - D(\nabla_x F)(x^*, u^*) \big). \tag{21}$$

*But since the sequence $(\alpha_k)_{k \in \mathbb{N}}$ is bounded and $D(\nabla_x F)$ is continuous, for any $\varepsilon > 0$, we can easily find $\delta > 0$ independent of $k \in \mathbb{N}$ such that $\|\mathcal{B}_k(x, u^*) - \mathcal{B}_k(x^*, u^*)\| < \varepsilon$ whenever $\|x - x^*\| < \delta$. Moreover, we also find from (21) that, for any sequence $(x^{(k)})_{k \in \mathbb{N}}$ converging to $x^*$, the sequence $\mathcal{B}_k(x^{(k)}, u^*) - \mathcal{B}_k(x^*, u^*)$ converges to 0 therefore satisfying our Assumption 2.2 for Gradient Descent. Note that, when additionally, the step size sequence converges, Assumption A.2 of Beck (1994) is also satisfied.*

We similarly define the notion of uniformly Lipschitz continuous sequence of functions.

**Definition A.3** (Uniform Lipschitz Continuity). A sequence of functions $(\mathcal{B}_k : \mathcal{X} \times \Omega \to \mathcal{X})_{k \in \mathbb{N}}$ with values $\mathcal{B}_k(x, u)$ is Lipschitz continuous in $x$ uniformly in $u$ and $k$, if for all $u \in \Omega$ and $k \in \mathbb{N}$, there exists $L \in \mathbb{R}$, such that, $\mathcal{B}_k(\cdot, u)$ is $L$-Lipschitz continuous. That is, for all $x, x^* \in \mathcal{X}$,

$$\|\mathcal{B}_k(x, u) - \mathcal{B}_k(x^*, u)\| \leq L\|x - x^*\|.$$

**Example A.4.** *Consider the setting of Example A.2 where $D(\nabla_x f)$ is additionally $M$-Lipschitz continuous. Clearly, from (21), for every $k \in \mathbb{N}$ and $u \in \mathcal{U}$, $\mathcal{B}_k(\cdot, u)$ is $\bar{\alpha}M$-Lipschitz continuous. Therefore, $(\mathcal{B}_k)_{k \in \mathbb{N}}$ is Lipschitz continuous in $x$ uniformly in $u$, and $k$ and for any sequence $(x^{(k)})_{k \in \mathbb{N}}$ converging to $x^*$, Gradient Descent with bounded step size sequence satisfies Assumption 2.3 because*

$$\|\mathcal{B}_k(x^{(k)}, u) - \mathcal{B}_k(x^*, u)\| \leq \bar{\alpha}M\|x^{(k)} - x^*\|.$$

*Finally, Assumption A.3 is satisfied when the step size sequence converges linearly which does not happen in general when employing line-search strategies for Gradient Descent.*

## A.2. Matrix Analysis

This section provides a few preliminary results on Matrix Analysis. We start by recalling a classical result from Linear Algebra which forms the foundation for the implicit differentiation applied to ($\mathcal{R}_e$) when combined with the Implicit Function Theorem (Dontchev & Rockafellar, 2009, Theorem 1B.1).

**Lemma A.5.** *For any linear operator $B\colon \mathcal{X} \to \mathcal{X}$ with $\rho(B) < 1$, the linear operator $I - B$ is invertible.*

We now provide some results which will be used in the proofs of convergence of automatic differentiation of the fixed-point iterations. In particular, we provide convergence guarantees of sequences generated through linear iterative procedures including (26), and (22) which bear strong resemblance with ($\mathcal{DR}$) and ($\mathcal{DR}_k$) (see the proofs of Theorems A.12 and 2.5 for a more detailed comparison). These convergence results are mainly derived from Polyak (1987, Section 2.1.2, Theorem 1), Riis (2020, Proposition 2.7), and Mehmood & Ochs (2022, Theorem 8). The following theorem will be used to prove convergence results in Section 2.

**Theorem A.6.** *Let $(B_k)_{k\in\mathbb{N}_0}$, $(C_k)_{k\in\mathbb{N}_0}$ and $(d^{(k)})_{k\in\mathbb{N}_0}$ be sequences in $\mathcal{L}(\mathcal{X},\mathcal{X})$, $\mathcal{L}(\mathcal{X},\mathcal{X})$ and $\mathcal{X}$ respectively such that $C_k \to 0$, $d^{(k)} \to 0$ and $\rho := \limsup_{k\to\infty} \|B_k\| < 1$ where $\|\cdot\|$ is a norm on $\mathcal{L}(\mathcal{X},\mathcal{X})$ induced by some vector norm. Then the sequence $(e^{(k)})_{k\in\mathbb{N}_0}$, with $e^{(0)} \in \mathcal{X}$, generated by*

$$e^{(k+1)} := B_k e^{(k)} + C_k e^{(k)} + d^{(k)}, \tag{22}$$

*converges to $0$. The convergence is linear when $C_k$ and $d^{(k)}$ converge linearly with rates $q_C$ and $q_d$ respectively. In fact, for all $\varepsilon \in (0, 1 - \rho)$, there exist $K \in \mathbb{N}$, such that for all $k \geq K$, we have*

$$\|e^{(k)}\| \leq \xi_1 (\rho + \varepsilon)^{k-K} + \xi_2 (k - K) q^{k-K}, \tag{23}$$

*where*

$$\begin{aligned}
q &:= \max(\rho + \varepsilon, q_C, q_d) \\
\xi_1 &:= \|e^{(K)}\| \\
\xi_2 &:= \|C_K e^{(K)} + d^{(K)}\|/q.
\end{aligned} \tag{24}$$

*Proof.* From classical analysis, given $\varepsilon \in (0, 1 - \rho)$, there exists $K \in \mathbb{N}$ such that $\|B_k\| < \rho + \varepsilon/2$ for all $k \geq K$. Also, with $K$ large enough we have $\|C_k e^{(k)}\|/\|e^{(k)}\| \leq \varepsilon/2$ since $C_k \to 0$. Therefore, from (22), we get

$$\begin{aligned}
\|e^{(k+1)}\| &\leq (\rho + \epsilon/2)\|e^{(k)}\| + \|C_k e^{(k)}\| + \|d^{(k)}\| \\
&\leq (\rho + \epsilon)\|e^{(k)}\| + \|d^{(k)}\|.
\end{aligned}$$

The convergence of $e^{(k)}$ then follows from the arguments following Equation 34 in the proof of Theorem 8 in Mehmood & Ochs (2022).

For the rate of convergence, we note that $\varepsilon^{(k)} := C_k e^{(k)} + d^{(k)}$ converges linearly to $0$ with rate $\max(q_C, q_d)$ and for $K \in \mathbb{N}$ sufficiently large we can write $\|\varepsilon^{(k)}\| \leq \max(q_C, q_d)^{k-K}\|\varepsilon^{(K)}\|$ for all $k \geq K$. Thus, for any $k \geq K$, we expand the expression $e^{(k+1)} = B_k e^{(k)} + \varepsilon^{(k)}$ to obtain

$$e^{(k+1)} = B_{k:K} e^{(K)} + \sum_{i=K}^{k} B_{k:i+1} \varepsilon^{(i)}$$

where $B_{k:i}$ denotes the ordered product $B_k B_{k-1} \cdots B_{i+1} B_i$ when $i \leq k$ and identity operator $I$ when $i > k$. Observe that, for any $\varepsilon \in (0, 1 - \rho)$ and $K \in \mathbb{N}$ large enough, $\|B_k\| < \rho + \varepsilon$ for all $k \geq K$ and therefore for any $K \leq i \leq k + 1$, we

have $\|B_{k:i}\| \le (\rho + \varepsilon)^{k-i+1}$. Therefore, setting $q := \max(\rho + \varepsilon, q_C, q_d)$, we end up with

$$
\begin{aligned}
\|\boldsymbol{e}^{(k+1)}\| &\le \|B_{k:K}\|\|\boldsymbol{e}^{(K)}\| + \sum_{i=K}^{k} \|B_{k:i+1}\|\|\boldsymbol{\varepsilon}^{(i)}\| \\
&\le \|\boldsymbol{e}^{(K)}\|(\rho + \varepsilon)^{k-K+1} + \sum_{i=K}^{k}(\rho + \varepsilon)^{k-i}\|\boldsymbol{\varepsilon}^{(K)}\| \max(q_C, q_d)^{i-K} \\
&\le \|\boldsymbol{e}^{(K)}\|(\rho + \varepsilon)^{k-K+1} + \|\boldsymbol{\varepsilon}^{(K)}\| \sum_{i=K}^{k} q^{k-K} \\
&\le \|\boldsymbol{e}^{(K)}\|(\rho + \varepsilon)^{k-K+1} + \|\boldsymbol{\varepsilon}^{(K)}\|(k-K+1)q^{k-K} \\
&= \xi_1(\rho + \varepsilon)^{k-K+1} + \xi_2(k-K+1)q^{k-K+1} ,
\end{aligned}
\tag{25}
$$

where $\xi_1 := \|\boldsymbol{e}^{(K)}\|$, and $\xi_2 := \|\boldsymbol{\varepsilon}^{(K)}\|/q$. $\qquad\square$

The result below is a direct consequence of the above theorem and will find its use when proving Corollary 2.9.

**Corollary A.7.** *The conclusion of Theorem A.6 also holds when the assumptions on the sequence $(B_k)_{k \in \mathbb{N}_0}$ are replaced with $B_k \to B \in \mathcal{L}(\mathcal{X}, \mathcal{X})$ and $\rho(B) < 1$.*

*Proof.* This is a special case of Theorem A.6 because for any $\delta \in (0, 1 - \rho(B))$, there exists a norm on $\mathcal{L}(\mathcal{X}, \mathcal{X})$ induced by some vector norm, both denoted by $\|\cdot\|_\delta$, such that $\|B\|_\delta \le \rho(B) + \delta$ and $\rho := \lim_{k \to \infty} \|B_k\|_\delta = \|B\|_\delta \le \rho(B) + \delta < 1$. $\qquad\square$

The following result is a special case of the setting of Corollary A.7 and is straightforward to show.

**Corollary A.8.** *The conclusion of Theorem A.6 also holds when the assumptions on the sequence $(B_k)_{k \in \mathbb{N}_0}$ are replaced with $B_k := B \in \mathcal{L}(\mathcal{X}, \mathcal{X})$ for all $k \in \mathbb{N}$ and $\rho(B) < 1$.*

Using Corollary A.8, we can prove the following statement which will be useful in the convergence proofs of Section A.3.

**Theorem A.9.** *Let $(B_k)_{k \in \mathbb{N}_0}$ and $(\boldsymbol{b}^{(k)})_{k \in \mathbb{N}_0}$ be sequences in $\mathcal{L}(\mathcal{X}, \mathcal{X})$ and $\mathcal{X}$ with limits $B$ and $\boldsymbol{b}$, respectively. If $\rho = \rho(B) < 1$, the sequence $(\boldsymbol{x}^{(k)})_{k \in \mathbb{N}_0}$, with $\boldsymbol{x}^{(0)} \in \mathcal{X}$, generated by*

$$
\boldsymbol{x}^{(k+1)} := B_k \boldsymbol{x}^{(k)} + \boldsymbol{b}^{(k)} ,
\tag{26}
$$

*converges to $\boldsymbol{x} := (I - B)^{-1}\boldsymbol{b}$. The convergence is linear when $(B_k)_{k \in \mathbb{N}_0}$ and $(\boldsymbol{b}^{(k)})_{k \in \mathbb{N}_0}$ converge linearly with rates $q_B$ and $q_{\boldsymbol{b}}$, respectively. In fact, for all $\delta \in (0, 1 - \rho)$, there exist $C_1(\delta)$, $C_2(\delta)$ and $K \in \mathbb{N}$, such that for all $k \ge K$, we have*

$$
\|\boldsymbol{x}^{(k)} - \boldsymbol{x}\| \le C_1(\delta)(k - K)q^{k-K} + C_2(\delta)(\rho + \delta)^{k-K} ,
\tag{27}
$$

*where $q := \max(\rho + \delta, q_B, q_{\boldsymbol{b}})$.*

*Proof.* The expression $\boldsymbol{x} = (I - B)^{-1}\boldsymbol{b}$ is well-defined thanks to Lemma A.5 and solves the linear equation $\boldsymbol{x} = B\boldsymbol{x} + \boldsymbol{b}$ for $\boldsymbol{x}$. By setting $\boldsymbol{e}^{(k)} := \boldsymbol{x}^{(k)} - \boldsymbol{x}$, $C_k := B_k - B$, and $\boldsymbol{d}^{(k)} := (B_k - B)\boldsymbol{x} + \boldsymbol{b}^{(k)} - \boldsymbol{b}$, we obtain,

$$
\begin{aligned}
\boldsymbol{e}^{(k+1)} &= (B_k \boldsymbol{x}^{(k)} + \boldsymbol{b}^{(k)}) - (B\boldsymbol{x} + \boldsymbol{b}) \\
&= B\boldsymbol{e}^{(k)} + C_k \boldsymbol{e}^{(k)} + \boldsymbol{d}^{(k)} .
\end{aligned}
$$

Since the above recursion matches (22) and the setting of Corollary A.8 applies, the result follows. $\qquad\square$

### A.3. Differentiation of Iteration-Dependent Algorithms: Classical Results

In this section, we briefly recap the results of Beck (1994) in a way that aligns with those of Section 2, allowing for a clearer comparison between the two sets of results. We first lay down the assumptions on the sequence of mappings $\mathcal{A}_k$ in $(\mathcal{R}_k)$ and the mapping $\mathcal{A}$ in $(\mathcal{R}_e)$ which will ensure the convergence of the sequence $D\boldsymbol{x}^{(k)}(\boldsymbol{u})$ generated by $(\mathcal{DR}_k)$. The main requirement in the work of Beck (1994) is that of the pointwise convergence of the sequence $D\mathcal{A}_k$ to $D\mathcal{A}$.

A.3.1. PROBLEM SETTING

Given $\boldsymbol{u}^* \in \mathcal{X}$, we assume that $\boldsymbol{x}^*$ solves $(\mathcal{R}_e)$ for $\boldsymbol{x}$ with $\boldsymbol{u} = \boldsymbol{u}^*$ and is the desired limit of $\boldsymbol{x}^{(k)}(\boldsymbol{u}^*)$ generated by $(\mathcal{R}_k)$. Furthermore, we assume that for all $k$, $\mathcal{A}_k$ and $\mathcal{A}$ are $C^1$-smooth near $(\boldsymbol{x}^*, \boldsymbol{u}^*)$ and the contraction property holds for $D_{\boldsymbol{x}}\mathcal{A}$ at $(\boldsymbol{x}^*, \boldsymbol{u}^*)$. In particular, given $(\boldsymbol{x}^{(0)}, \boldsymbol{x}^*, \boldsymbol{u}^*) \in \mathcal{X} \times \mathcal{X} \times \mathcal{U}$, $V \in \mathcal{N}_{(\boldsymbol{x}^*, \boldsymbol{u}^*)}$, $(\mathcal{A}_k)_{k \in \mathbb{N}_0}$, and $\mathcal{A}$ we assume that the following assumption holds.

**Assumption A.1.** (i) $\mathcal{A}_k|_V$ and $\mathcal{A}|_V$ are $C^1$-smooth for all $k$,

(ii) $\boldsymbol{x}^* = \mathcal{A}(\boldsymbol{x}^*, \boldsymbol{u}^*)$,

(iii) $\rho(D_{\boldsymbol{x}}\mathcal{A}(\boldsymbol{x}^*, \boldsymbol{u}^*)) < 1$, and

(iv) $\boldsymbol{x}^{(k)}(\boldsymbol{u}^*)$ generated by $(\mathcal{R}_k)$ has limit $\boldsymbol{x}^*$ such that $(\boldsymbol{x}^{(k)}(\boldsymbol{u}^*), \boldsymbol{u}^*) \in V$ for all $k \in \mathbb{N}$.

A.3.2. IMPLICIT DIFFERENTIATION

From Assumption A.1, Lemma A.5, and (Dontchev & Rockafellar, 2009, Theorem 1B.1), we obtain the Implicit Function Theorem for $(\mathcal{R}_e)$.

**Theorem A.10** (Implicit Function Theorem). *For some $(\boldsymbol{x}^*, \boldsymbol{u}^*)$, let $\mathcal{A}$ be $C^1$-smooth near $(\boldsymbol{x}^*, \boldsymbol{u}^*)$, $\boldsymbol{x}^* = \mathcal{A}(\boldsymbol{x}^*, \boldsymbol{u}^*)$ and $\rho(D_{\boldsymbol{x}}\mathcal{A}(\boldsymbol{x}^*, \boldsymbol{u}^*)) < 1$. Then $\exists\, U \in \mathcal{N}_{\boldsymbol{u}^*}$ and a $C^1$-smooth mapping $\psi\colon U \to \mathcal{X}$ such that $\forall \boldsymbol{u} \in U$, $\psi(\boldsymbol{u}) = \mathcal{A}(\psi(\boldsymbol{u}), \boldsymbol{u})$, and*

$$D\psi(\boldsymbol{u}) = \left(I - D_{\boldsymbol{x}}\mathcal{A}(\psi(\boldsymbol{u}), \boldsymbol{u})\right)^{-1} D_{\boldsymbol{u}}\mathcal{A}(\psi(\boldsymbol{u}), \boldsymbol{u}) \,. \tag{28}$$

A.3.3. AUTOMATIC DIFFERENTIAION

As stated in Section A.2, the update procedure to generate the derivative iterates in $(\mathcal{DR}_k)$ takes after the iterative process defined by (26) in Theorem A.9. That is, for some $\boldsymbol{u}^* \in \mathcal{U}$ and $\dot{\boldsymbol{u}} \in \mathcal{U}$, if we set $B_k := D_{\boldsymbol{x}}\mathcal{A}_k(\boldsymbol{x}^{(k)}(\boldsymbol{u}^*), \boldsymbol{u}^*)$ and $\boldsymbol{b}^{(k)} := D_{\boldsymbol{u}}\mathcal{A}_k(\boldsymbol{x}^{(k)}(\boldsymbol{u}^*), \boldsymbol{u}^*)\dot{\boldsymbol{u}}$, the resulting sequence is $\boldsymbol{y}^{(k)} = D\boldsymbol{x}^{(k)}(\boldsymbol{u}^*)\dot{\boldsymbol{u}}$. Similarly, the limit $\boldsymbol{y}^* := (I - B)^{-1}\boldsymbol{b}$ of the sequence $\boldsymbol{y}^{(k)}$ generated by (26) matches $D\psi(\boldsymbol{u}^*)\dot{\boldsymbol{u}}$ from (28), if we set $B := D_{\boldsymbol{x}}\mathcal{A}(\psi(\boldsymbol{u}^*), \boldsymbol{u}^*)$ and $\boldsymbol{b} := D_{\boldsymbol{u}}\mathcal{A}(\psi(\boldsymbol{u}^*), \boldsymbol{u}^*)\dot{\boldsymbol{u}}$. One way to prove the convergence of $D\boldsymbol{x}^{(k)}(\boldsymbol{u}^*)\dot{\boldsymbol{u}}$ to $D\psi(\boldsymbol{u}^*)\dot{\boldsymbol{u}}$ is by asserting that $D\mathcal{A}_k(\boldsymbol{x}^{(k)}(\boldsymbol{u}^*), \boldsymbol{u}^*)$ converges to $D\mathcal{A}(\psi(\boldsymbol{u}^*), \boldsymbol{u}^*)$.

**Assumption A.2.** The sequence $(D\mathcal{A}_k(\boldsymbol{x}^{(k)}(\boldsymbol{u}^*), \boldsymbol{u}^*))_{k \in \mathbb{N}_0}$ converges to $D\mathcal{A}(\boldsymbol{x}^*, \boldsymbol{u}^*)$.

*Remark* A.11. When the sequence of functions $(D\mathcal{A}_k)_{k \in \mathbb{N}}$ is equicontinuous at $(\boldsymbol{x}^*, \boldsymbol{u}^*)$ and has a pointwise limit $D\mathcal{A}$ near $(\boldsymbol{x}^*, \boldsymbol{u}^*)$, then Assumption A.2 naturally holds thanks to (5).

The main result of Beck (1994) can then be stated below.

**Theorem A.12.** *Let $(\boldsymbol{x}^{(0)}, \boldsymbol{x}^*, \boldsymbol{u}^*)$ be such that Assumptions A.1 and A.2 are satisfied by $\mathcal{A}_k$ and $\mathcal{A}$. Then the sequence $(D\boldsymbol{x}^{(k)}(\boldsymbol{u}^*))_{k \in \mathbb{N}_0}$ generated by $(\mathcal{DR}_k)$ converges to $D\psi(\boldsymbol{u}^*)$.*

*Proof.* The proof is a direct consequence of Assumptions A.1 and A.2, Theorem A.9 and the arguments following Theorem A.9. □

For linear convergence we require a stronger assumption on $D\mathcal{A}_k$. That is, it is linear convergence as given below.

**Assumption A.3.** The sequence $(\boldsymbol{x}^{(k)}(\boldsymbol{u}^*))_{k \in \mathbb{N}_0}$ converges linearly to $\boldsymbol{x}^*$ and

$$D\mathcal{A}_k(\boldsymbol{x}^{(k)}(\boldsymbol{u}^*), \boldsymbol{u}^*) - D\mathcal{A}(\boldsymbol{x}^*, \boldsymbol{u}^*) = \mathcal{O}(\boldsymbol{x}^{(k)}(\boldsymbol{u}^*) - \boldsymbol{x}^*) \,. \tag{29}$$

**Theorem A.13.** *Let $(\boldsymbol{x}^{(0)}, \boldsymbol{x}^*, \boldsymbol{u}^*)$ be such that Assumptions A.1 and A.3 are satisfied by $\mathcal{A}_k$ and $\mathcal{A}$. Then the sequence $(D\boldsymbol{x}^{(k)}(\boldsymbol{u}^*))_{k \in \mathbb{N}_0}$ generated by $(\mathcal{DR}_k)$ converges linearly to $D\psi(\boldsymbol{u}^*)$. In particular, when the sequence $(\boldsymbol{x}^{(k)})_{k \in \mathbb{N}_0}$ converges with rate $q_{\boldsymbol{x}} < 1$, then for all $\delta \in (0, 1 - \rho)$, there exist $C_1(\delta), C_2(\delta)$ and $K \in \mathbb{N}$, such that for all $k \geq K$, we have*

$$\|D\boldsymbol{x}^{(k)}(\boldsymbol{u}^*) - D\psi(\boldsymbol{u}^*)\| \leq C_1(\delta)(k - K)q^{k-K} + C_2(\delta)(\rho + \delta)^{k-K} \,, \tag{30}$$

*where $\rho := \rho(D_{\boldsymbol{x}}\mathcal{A}(\boldsymbol{x}^*, \boldsymbol{u}^*))$ and $q := \max(\rho + \delta, q_{\boldsymbol{x}})$.*

*Proof.* Assumption A.3 asserts that the sequences $D_{\boldsymbol{x}}\mathcal{A}_k(\boldsymbol{x}^{(k)}(\boldsymbol{u}^*),\boldsymbol{u}^*)$ and $D_{\boldsymbol{u}}\mathcal{A}_k(\boldsymbol{x}^{(k)}(\boldsymbol{u}^*),\boldsymbol{u}^*)\dot{\boldsymbol{u}}$ respectively converge to $D_{\boldsymbol{x}}\mathcal{A}(\boldsymbol{x}^*,\boldsymbol{u}^*)$ and $D_{\boldsymbol{u}}\mathcal{A}(\boldsymbol{x}^*,\boldsymbol{u}^*)\dot{\boldsymbol{u}}$ for any $\dot{\boldsymbol{u}} \in \mathcal{U}$. Furthermore, the rate of convergence of the two sequences is linear and is the same as that of $\boldsymbol{x}^{(k)}$, that is, $q_{\boldsymbol{x}}$. Therefore, the proof follows by simply invoking the second part of Theorem A.9. $\qquad\square$

*Remark* A.14. Assumption A.3 is not practical and therefore we rarely see any application of the above result in practice. For instance, for gradient descent with line search, it requires linear convergence of the step size sequence which does not hold in general.

### A.4. Riemannian Geometry

We briefly recall a few definitions and results from Riemannian Geometry. For further details, the reader is referred to standard texts such as Lee (2003); Chavel (2006).

**Definition A.15** (Manifold, Tangent and Normal Spaces). For any $k \in \mathbb{N}$, we say that $\mathcal{M} \subset \mathcal{X}$ is a $C^k$-smooth $m$-dimensional submanifold of $\mathcal{X}$, if for every $\boldsymbol{x} \in \mathcal{M}$ there exist an open set $X \subset \mathcal{X}$ and a $C^k$-smooth map $\Phi\colon X \to \mathbb{R}^{N-m}$ such that $\boldsymbol{x} \in X$, the derivative $D\Phi(\boldsymbol{x})$ is surjective, and $X \cap \mathcal{M} = \Phi^{-1}(0) = \{\boldsymbol{y} \in X \ :\ \Phi(\boldsymbol{y}) = 0\}$. We call $\Phi$, the local defining function of $\mathcal{M}$ at $\boldsymbol{x}$. We say that $\boldsymbol{v} \in \mathcal{X}$ is a tangent vector of $\mathcal{M}$ at $\boldsymbol{x}$ if there exist $\epsilon > 0$ and a $C^1$-smooth curve $\gamma\colon (-\epsilon, \epsilon) \to \mathcal{M}$ on $\mathcal{M}$ with $\gamma(0) = \boldsymbol{x}$ and $\dot{\gamma}(0) = \boldsymbol{v}$. The set of all tangent vectors of $\mathcal{M}$ at $\boldsymbol{x}$ constitute $T_{\boldsymbol{x}}\mathcal{M}$, the tangent space of $\mathcal{M}$ at $\boldsymbol{x}$. We define the normal space of $\mathcal{M}$ at $\boldsymbol{x}$ by $N_{\boldsymbol{x}}\mathcal{M} := (T_{\boldsymbol{x}}\mathcal{M})^{\perp}$, the orthogonal complement of $T_{\boldsymbol{x}}\mathcal{M}$.

For any $\boldsymbol{x} \in \mathcal{M}$ and a local defining function $\Phi$ for $\mathcal{M}$ at $\boldsymbol{x}$, we have $T_{\boldsymbol{x}}\mathcal{M} = \ker D\Phi(\boldsymbol{x})$.

**Definition A.16** (Riemannian Gradient). For any $k \in \mathbb{N}$, let $\mathcal{M} \subset \mathcal{X}$ be a $C^k$-smooth manifold, $f\colon \mathcal{M} \to \mathbb{R}$ be a function and $\boldsymbol{x} \in \mathcal{M}$. We say that $f$ is $C^k$-smooth at $\boldsymbol{x}$ if there exist a neighbourhood $X \subset \mathcal{X}$ of $\boldsymbol{x}$ and a $C^k$-smooth function $\tilde{f}\colon X \to \mathbb{R}$ such that $\tilde{f}$ agrees with $f$ on $\mathcal{M} \cap X$. In this case, we call $\tilde{f}$ a smooth extension of $f$ around $\boldsymbol{x}$. We call $\nabla_{\mathcal{M}}f(\boldsymbol{x}) \in T_{\boldsymbol{x}}\mathcal{M}$, the Riemannian gradient of $f$ at $\boldsymbol{x}$ if for all $\boldsymbol{v} \in T_{\boldsymbol{x}}\mathcal{M}$, $\langle \nabla_{\mathcal{M}}f(\boldsymbol{x}), \boldsymbol{v} \rangle = (f \circ \gamma)'(0)$, where $\gamma\colon (-\epsilon, \epsilon) \to \mathcal{M}$ is any $C^1$-curve with $\gamma(0) = \boldsymbol{x}$ and $\dot{\gamma}(0) = \boldsymbol{v}$.

The Riemannian Gradient of $f$ can also be expressed in terms of the gradient of the smooth extension $\tilde{f}$ of $f$ by

$$\nabla_{\mathcal{M}}f(\boldsymbol{x}) = \Pi(\boldsymbol{x})\nabla\tilde{f}(\boldsymbol{x}).$$

Note that this gradient neither depends on the choice of the curve $\gamma$ nor the smooth extension $\tilde{f}$.

**Definition A.17** (Riemannian Hessian). Let $\mathcal{M} \subset \mathcal{X}$ be a $C^2$-smooth manifold, $f\colon \mathcal{M} \to \mathbb{R}$ be a $C^2$-smooth function and $\boldsymbol{x} \in \mathcal{M}$. We call $\nabla^2_{\mathcal{M}}f(\boldsymbol{x})\colon T_{\boldsymbol{x}}\mathcal{M} \to T_{\boldsymbol{x}}\mathcal{M}$, the Riemannian Hessian of $f$ at $\boldsymbol{x}$ if for all $\boldsymbol{v} \in T_{\boldsymbol{x}}\mathcal{M}$, $\langle \nabla^2_{\mathcal{M}}f(\boldsymbol{x})\boldsymbol{v}, \boldsymbol{v} \rangle = (f \circ \gamma)''(0)$, where $\gamma\colon (-\epsilon, \epsilon) \to \mathcal{M}$ is any $C^1$-curve with $\gamma(0) = \boldsymbol{x}$ and $\dot{\gamma}(0) = \boldsymbol{v}$.

We can similarly express the Riemannian Hessian $\nabla^2_{\mathcal{M}}f(\boldsymbol{x}) \in \mathcal{L}(T_{\boldsymbol{x}}\mathcal{M}, T_{\boldsymbol{x}}\mathcal{M})$ by using the smooth extension $\tilde{f}$, that is,

$$\nabla^2_{\mathcal{M}}f(\boldsymbol{x}) = \Pi(\boldsymbol{x})\nabla^2\tilde{f}(\boldsymbol{x}) + \mathfrak{W}_{\boldsymbol{x}}(\cdot, \Pi^{\perp}(\boldsymbol{x})\nabla\tilde{f}(\boldsymbol{x})), \tag{31}$$

where the mapping $\mathfrak{W}_{\boldsymbol{x}}(\cdot, \boldsymbol{w}) \in \mathcal{L}(T_{\boldsymbol{x}}\mathcal{M}, T_{\boldsymbol{x}}\mathcal{M})$ for $\boldsymbol{w} \in N_{\boldsymbol{x}}\mathcal{M}$ is called the Weingarten map. It is defined by

$$\boldsymbol{v} \mapsto \mathfrak{W}_{\boldsymbol{x}}(\boldsymbol{v}, \boldsymbol{w}) := -\Pi(\boldsymbol{x})\mathrm{d}W[\boldsymbol{v}],$$

where $W$ is a local extension of $\boldsymbol{w}$ to a normal vector field on $\mathcal{M}$. $\mathfrak{W}_{\boldsymbol{x}}(\cdot, \boldsymbol{w})$ is independent of the choice of normal field $W$ (Chavel, 2006, Proposition II.2.1) and caters for the change of the tangent space as we move away from $\boldsymbol{x}$. It vanishes when the manifold is affine, that is, $\mathcal{M} = \boldsymbol{x} + T_{\boldsymbol{x}}\mathcal{M}$ or linear, that is, $\mathcal{M} = T_{\boldsymbol{x}}\mathcal{M}$. The Hessian expression then reduces to $\nabla^2_{\mathcal{M}}f(\boldsymbol{x}) = \Pi(\boldsymbol{x})\nabla^2\tilde{f}(\boldsymbol{x})$.

### A.5. Partial Smoothness

Introduced by Lewis (2002), partial smoothness describes a broad class of non-smooth functions, including many commonly used loss and regularization functions in Machine Learning. It extends ideas like active set methods which were originally developed for smooth constrained optimization to a more general setting that includes non-smooth functions, by identifying smooth behavior along certain submanifolds.

**Definition A.18** (Partial Smoothness). Let $f\colon \mathcal{X} \to \overline{\mathbb{R}}$ be proper and lower semi-continuous and $\mathcal{M} \subset \mathcal{X}$ be a set. We say that $f$ is partly smooth at a point $\boldsymbol{x} \in \mathcal{M}$ relative to $\mathcal{M}$ if the following conditions hold:

(i) (Regularity:) $f$ is regular at every point close to $\boldsymbol{x}$ and $\partial f(\boldsymbol{x}) \neq \emptyset$.

(ii) (Smoothness:) $\mathcal{M}$ is a $C^2$-smooth manifold and $f|_{\mathcal{M}}$ is $C^2$-smooth around $\boldsymbol{x}$.

(iii) (Sharpness:) $N_{\boldsymbol{x}}\mathcal{M} = \operatorname{par} \partial f(\boldsymbol{x})$.

(iv) (Continuity:) $\partial f$ is continuous at $\boldsymbol{x}$ relative to $\mathcal{M}$.

We call $f$ partly smooth relative to $\mathcal{M}$ if $f$ is partly smooth at every $\boldsymbol{x} \in \mathcal{M}$ relative to $\mathcal{M}$.

## B. Proofs of Section 2

### B.1. Proof of Lemma 2.2

*Proof.* From Assumption 2.1(iii), given $\varepsilon \in (0, 1 - \rho)$, there exists $K \in \mathbb{N}$ such that $\rho(D_{\boldsymbol{x}}\mathcal{A}_k(\boldsymbol{x}^*, \boldsymbol{u}^*)) \leq \|D_{\boldsymbol{x}}\mathcal{A}_k(\boldsymbol{x}^*, \boldsymbol{u}^*)\| < \rho + \varepsilon$ for all $k \geq K$, where $\rho := \limsup_{k \to \infty} \|D_{\boldsymbol{x}}\mathcal{A}_k(\boldsymbol{x}^*, \boldsymbol{u}^*)\|$. The existence of the neighbourhood $U$ and the $C^1$-smooth mapping $\psi \colon U \to \mathcal{X}$ is guaranteed by Theorem A.10 applied to $\mathcal{A}_n$ for some $n \geq K$, thanks to Assumption 2.1 ((i)–(ii)). In particular, for all $\boldsymbol{u} \in U$, $\psi(\boldsymbol{u}) = \mathcal{A}_n(\psi(\boldsymbol{u}), \boldsymbol{u})$ and $D\psi(\boldsymbol{u})$ is given by (28) with $\mathcal{A}$ replaced by $\mathcal{A}_n$. From Assumption 2.1(ii), for any $k \geq K$ and $\boldsymbol{u} \in U$, we have $\psi(\boldsymbol{u}) = \mathcal{A}_n(\psi(\boldsymbol{u}), \boldsymbol{u}) = \mathcal{A}_k(\psi(\boldsymbol{u}), \boldsymbol{u})$ and with $U$ possibly reduced, $\rho(D_{\boldsymbol{x}}\mathcal{A}_k(\psi(\boldsymbol{u}), \boldsymbol{u})) < 1$. Therefore, by using the Chain rule and Lemma A.5, we obtain the expression in (4). □

### B.2. Proof of Theorem 2.5

*Proof.* From Lemma 2.2 and Remark 2.3, there exists $K \in \mathbb{N}$ such that for any $k \geq K$, the fixed-point mapping $\psi_k$ of $\mathcal{A}_k(\cdot, \boldsymbol{u})$ is $C^1$-smooth near $\boldsymbol{u}^*$ and $D\psi_k(\boldsymbol{u}^*) = X_*$ (see Assumption 2.1(ii)). We denote $\boldsymbol{x}^{(k)} := \boldsymbol{x}^{(k)}(\boldsymbol{u}^*)$ for simplicity and define

$$
\begin{aligned}
\boldsymbol{e}^{(k)} &:= \left( D\boldsymbol{x}^{(k)}(\boldsymbol{u}^*) - X_* \right) \dot{\boldsymbol{u}} \\
B_k &:= D_{\boldsymbol{x}}\mathcal{A}_k(\boldsymbol{x}^*, \boldsymbol{u}^*) \\
C_k &:= D_{\boldsymbol{x}}\mathcal{A}_k(\boldsymbol{x}^{(k)}, \boldsymbol{u}^*) - D_{\boldsymbol{x}}\mathcal{A}_k(\boldsymbol{x}^*, \boldsymbol{u}^*) \\
\boldsymbol{d}^{(k)} &:= \left( D\mathcal{A}_k(\boldsymbol{x}^{(k)}, \boldsymbol{u}^*) - D\mathcal{A}_k(\boldsymbol{x}^*, \boldsymbol{u}^*) \right) \begin{bmatrix} X_*\dot{\boldsymbol{u}} \\ \dot{\boldsymbol{u}} \end{bmatrix},
\end{aligned}
\tag{32}
$$

to obtain

$$
\begin{aligned}
\boldsymbol{e}^{(k+1)} &= \left( D_{\boldsymbol{x}}\mathcal{A}_k(\boldsymbol{x}^{(k)}, \boldsymbol{u}^*)D\boldsymbol{x}^{(k)}(\boldsymbol{u}^*)\dot{\boldsymbol{u}} + D_{\boldsymbol{u}}\mathcal{A}_k(\boldsymbol{x}^{(k)}, \boldsymbol{u}^*)\dot{\boldsymbol{u}} \right) - \\
&\quad \left( D_{\boldsymbol{x}}\mathcal{A}_k(\boldsymbol{x}^*, \boldsymbol{u}^*)X_*\dot{\boldsymbol{u}} + D_{\boldsymbol{u}}\mathcal{A}_k(\boldsymbol{x}^*, \boldsymbol{u}^*)\dot{\boldsymbol{u}} \right) \\
&= B_k \boldsymbol{e}^{(k)} + C_k \boldsymbol{z}^{(k)} + \boldsymbol{d}^{(k)} .
\end{aligned}
\tag{33}
$$

From Assumption 2.2 and the definitions of $C_k$ and $\boldsymbol{y}^{(k)}$, we note that $C_k \to 0$ and $\boldsymbol{d}^{(k)} \to 0$. Therefore, from Theorem A.6, $(D\boldsymbol{x}^{(k)}(\boldsymbol{u}^*)\dot{\boldsymbol{u}})_{k \in \mathbb{N}_0}$ converges to $X_*\dot{\boldsymbol{u}}$ for any $\dot{\boldsymbol{u}} \in \mathcal{U}$. □

### B.3. Proof of Theorem 2.7

*Proof.* From Assumption 2.3, we have $D\mathcal{A}_k(\boldsymbol{x}^{(k)}(\boldsymbol{u}^*), \boldsymbol{u}^*) - D\mathcal{A}_k(\boldsymbol{x}^*, \boldsymbol{u}^*) \to 0$ and the rate of convergence of the two sequences is $q_{\boldsymbol{x}}$. For any $\dot{\boldsymbol{u}} \in \mathcal{U}$, by using the definitions in (32), we note that both $C_k$ and $\boldsymbol{y}^{(k)}$ converge linearly with rate $q_{\boldsymbol{x}}$. Thus, invoking the second part of Theorem A.6, we obtain the convergence rate of $(D\boldsymbol{x}^{(k)}(\boldsymbol{u}^*)\dot{\boldsymbol{u}})_{k \in \mathbb{N}_0}$. In particular, from (23), we obtain the bound in (7). The expressions for $\xi_1$ and $\xi_2$ can be computed from (24) using $C_K$, $\boldsymbol{d}^{(K)}$ and $\boldsymbol{e}^{(K)}$ defined in (32). □

## C. Proofs of Section 3

### C.1. Proof of Theorem 3.4

The proof of Theorem 3.4 relies on the following preliminary result from set-valued analysis.

**Lemma C.1.** *Given a sequence of non-empty, closed, convex sets $C_k \subset \mathcal{X}$ which converges to $C \subset \mathcal{X}$ such that $\operatorname{aff} C_k$ also converges to $\operatorname{aff} C$. Let $\boldsymbol{x}^{(k)} \in C_k$ be a sequence with limit $\boldsymbol{x} \in \operatorname{ri} C$. Then there exists $K \in \mathbb{N}$ such that $\boldsymbol{x}^{(k)} \in \operatorname{ri} C_k$ for all $k \geq K$.*

*Proof.* Because $\boldsymbol{x}^{(k)} \to \boldsymbol{x}$ and $\boldsymbol{x} \in \operatorname{ri} C$, there exists a compact set $V$ and $K \in \mathbb{N}$ such that $\boldsymbol{x}^{(k)}, \boldsymbol{x} \in \operatorname{int} V \neq \emptyset$ for all $k \geq K$ and $V \cap \operatorname{aff} C \subset C$. Moreover, $\operatorname{aff} C_k \to \operatorname{aff} C$ implies $V \cap \operatorname{aff} C_k \to V \cap \operatorname{aff} C$ (Mosco, 1969, Lemma 1.4). Once we show that $V \cap \operatorname{aff} C_k \subset C_k$ eventually, we are done. Assume for contradiction, that there exists a subsequence $(C_{k_i})_{i \in \mathbb{N}}$ such that for all $i \in \mathbb{N}$, $\boldsymbol{y}^{(i)} \in V \cap \operatorname{aff} C_{k_i}$ and $\boldsymbol{y}^{(i)} \notin C_{k_i}$. The compactness of $V$ and convergence of $V \cap \operatorname{aff} C_{k_i}$ implies the existence of $\boldsymbol{y} \in V \cap \operatorname{aff} C$ such that $\boldsymbol{y}^{(i)} \to \boldsymbol{y}$, possibly through a subsequence. We define the bounded sequence $\boldsymbol{b}^{(i)} := \boldsymbol{a}^{(i)}/\|\boldsymbol{a}^{(i)}\|$ where $\boldsymbol{a}^{(i)} := \boldsymbol{y}^{(i)} - \operatorname{proj}_{C_{k_i}}(\boldsymbol{y}^{(i)})$ which lies in $\operatorname{par} C_{k_i}$ and has limit $0 \neq \boldsymbol{b} \in \operatorname{par} C$, again, possibly through a subsequence. For all $i \in \mathbb{N}$ and for all $\boldsymbol{w} \in C_{k_i}$,

$$
\begin{aligned}
\left\langle \boldsymbol{b}^{(i)}, \boldsymbol{w} - \boldsymbol{y}^{(i)} \right\rangle &= \left\langle \boldsymbol{b}^{(i)}, \boldsymbol{w} - \operatorname{proj}_{C_{k_i}}(\boldsymbol{y}^{(i)}) \right\rangle - \left\langle \boldsymbol{b}^{(i)}, \boldsymbol{y}^{(i)} - \operatorname{proj}_{C_{k_i}}(\boldsymbol{y}^{(i)}) \right\rangle \\
&= \left\langle \boldsymbol{b}^{(i)}, \boldsymbol{w} - \operatorname{proj}_{C_{k_i}}(\boldsymbol{y}^{(i)}) \right\rangle - \frac{1}{\|\boldsymbol{a}^{(i)}\|} \left\langle \boldsymbol{a}^{(i)}, \boldsymbol{a}^{(i)} \right\rangle \\
&= \left\langle \boldsymbol{b}^{(i)}, \boldsymbol{w} - \operatorname{proj}_{C_{k_i}}(\boldsymbol{y}^{(i)}) \right\rangle - \|\boldsymbol{a}^{(i)}\| \\
&\leq 0 \,,
\end{aligned}
$$

where the inequality follows because $\boldsymbol{a}^{(i)}$ and therefore $\boldsymbol{b}^{(i)}$ lie in $N_{C_{k_i}}(\operatorname{proj}_{C_{k_i}}(\boldsymbol{y}^{(i)}))$. The above inequality leads to $\langle \boldsymbol{b}, \boldsymbol{w} - \boldsymbol{y} \rangle \leq 0$ for all $\boldsymbol{w} \in C$. In other words, we obtain $0 \neq \boldsymbol{b} \in N_C(\boldsymbol{y}) \cap \operatorname{par} C$ which is a contradiction because $\boldsymbol{y} \in \operatorname{ri} C$. $\qquad\square$

*Remark C.2.* The condition $\operatorname{aff} C_k \to \operatorname{aff} C$ is necessary for the conclusion of Lemma C.1 to hold. To see why, we define the sequence of closed intervals $(C_k := [-1/k, 1/k])_{k \in \mathbb{N}}$ in $\mathbb{R}$ which converges to $C := \{0\}$. Consider the sequence $x_k := 1/k$ which converges to $x := 0$. Notice that, $\operatorname{aff} C_k = \mathbb{R}$ for all $k \in \mathbb{N}$, $\operatorname{aff} C = \{0\}$ and even though $x \in C = \operatorname{ri} C$, $\boldsymbol{x}^{(k)} \notin \operatorname{ri} C_k$ for all $k \in \mathbb{N}$.

*Proof of Theorem 3.4.* (i) From the convergence of $\boldsymbol{x}^{(k)} := \boldsymbol{x}^{(k)}(\boldsymbol{u}^*)$, Lemma 3.9 ensures that for $\delta > 0$ small enough, there exists $K \in \mathbb{N}$, such that $\boldsymbol{x}^{(k)} \in \mathcal{M} \cap B_\delta(\boldsymbol{x}^*)$ for all $k \geq K$. We define the maps $G \colon \mathcal{X} \times \Omega \times (0, 2/L) \to \mathcal{X}$ by $G(\boldsymbol{x}, \boldsymbol{u}, \alpha) = \boldsymbol{x} - \alpha \nabla_{\boldsymbol{x}} f(\boldsymbol{x}, \boldsymbol{u})$ and $H \colon \mathcal{X} \times \mathcal{X} \times \Omega \times (0, 2/L) \to \overline{\mathbb{R}}$ by

$$
H(\boldsymbol{y}, \boldsymbol{x}, \boldsymbol{u}, \alpha) := \alpha g(\boldsymbol{y}, \boldsymbol{u}) + \frac{1}{2} \|\boldsymbol{y} - \boldsymbol{x} + \alpha \nabla_{\boldsymbol{x}} f(\boldsymbol{x}, \boldsymbol{u})\|^2 \,,
$$

and note that $\boldsymbol{x}^{(k+1)} = \mathcal{A}_{\alpha_k}(\boldsymbol{x}^{(k)}, \boldsymbol{u}^*) = \operatorname{argmin}_{\boldsymbol{y}} H(\boldsymbol{y}, \boldsymbol{x}^{(k)}, \boldsymbol{u}^*, \alpha_k)$, which, from Fermat's rule, is equivalent to $0 \in \partial_{\boldsymbol{y}} H(\boldsymbol{x}^{(k+1)}, \boldsymbol{x}^{(k)}, \boldsymbol{u}^*, \alpha_k)$ or $\boldsymbol{\mu}^{(k)} \in \partial_{\boldsymbol{x}} F(\boldsymbol{x}^{(k+1)}, \boldsymbol{u}^*)$, where

$$
\boldsymbol{\mu}^{(k)} := \frac{1}{\alpha_k} \Big( G(\boldsymbol{x}^{(k)}, \boldsymbol{u}^*, \alpha_k) - G(\boldsymbol{x}^{(k+1)}, \boldsymbol{u}^*, \alpha_k) \Big) . \tag{34}
$$

Notice that $\boldsymbol{\mu}^{(k)} \to 0$ because $G(\cdot, \boldsymbol{u}^*, \alpha)$ is non-expansive when $\alpha \in (0, 2/L)$ and we have $\|\boldsymbol{\mu}^{(k)}\| \leq \frac{1}{\alpha}\|\boldsymbol{x}^{(k)} - \boldsymbol{x}^{(k+1)}\|$. Moreover, $\operatorname{aff} \partial_{\boldsymbol{x}} F(\boldsymbol{x}^{(k+1)}, \boldsymbol{u}^*) = \boldsymbol{\mu}^{(k)} + \operatorname{par} \partial_{\boldsymbol{x}} F(\boldsymbol{x}^{(k+1)}, \boldsymbol{u}^*) = \boldsymbol{\mu}^{(k)} + T_{\boldsymbol{x}^{(k+1)}} \mathcal{M}$ converges to $\operatorname{aff} \partial_{\boldsymbol{x}} F(\boldsymbol{x}^*, \boldsymbol{u}^*) = \operatorname{par} \partial_{\boldsymbol{x}} F(\boldsymbol{x}^*, \boldsymbol{u}^*) = 0 + T_{\boldsymbol{x}^*} \mathcal{M}$ (Mosco, 1969, Lemma 1.6). Therefore, from (ND) and Lemma C.1, we have $\boldsymbol{\mu}^{(k)} \in \operatorname{ri} \partial_{\boldsymbol{x}} F(\boldsymbol{x}^{(k+1)}, \boldsymbol{u}^*)$ or $0 \in \operatorname{ri} \partial_{\boldsymbol{y}} H(\boldsymbol{x}^{(k+1)}, \boldsymbol{x}^{(k)}, \boldsymbol{u}^*, \alpha_k)$, for all $k \in \mathbb{N}$ large enough. Since $H$ is partly smooth relative to $\mathcal{M} \times \mathcal{X} \times \Omega \times (0, 2/L)$ and $H(\cdot, \boldsymbol{x}, \boldsymbol{u}, \alpha)$ is strongly convex for all $(\boldsymbol{x}, \boldsymbol{u}, \alpha) \in \mathcal{X} \times \Omega \times (0, 2/L)$, and non-degeneracy condition holds, the $C^1$-smoothness of the update map near $(\boldsymbol{x}^{(k)}, \boldsymbol{u}^*, \alpha_k)$ follows by invoking Theorem 3.1. The $C^1$-smoothness of $\mathcal{A}_\alpha$ near $(\boldsymbol{x}^*, \boldsymbol{u}^*, \alpha)$ under given assumptions was shown in Mehmood & Ochs (2022, Corollary 32) for any $\alpha \in [\underline{\alpha}, \bar{\alpha}]$.

(ii) We define

$$
Q_k := \nabla_{\mathcal{M}}^2 H(\boldsymbol{x}^{(k+1)}, \boldsymbol{x}^{(k)}, \boldsymbol{u}^*, \alpha_k), \quad \tilde{Q}_k := \nabla_{\mathcal{M}}^2 H(\boldsymbol{x}^*, \boldsymbol{x}^*, \boldsymbol{u}^*, \alpha_k)
$$

$$
P_{\omega,k} := D_\omega \nabla_{\mathcal{M}} H(\boldsymbol{x}^{(k+1)}, \boldsymbol{x}^{(k)}, \boldsymbol{u}^*, \alpha_k), \quad \tilde{P}_{\omega,k} := D_\omega \nabla_{\mathcal{M}} H(\boldsymbol{x}^*, \boldsymbol{x}^*, \boldsymbol{u}^*, \alpha_k)
$$

$$
\Pi_k := \Pi(\boldsymbol{x}^{(k)}), \quad \Pi_* := \Pi(\boldsymbol{x}^*), \quad \Pi_k^\perp := \Pi^\perp(\boldsymbol{x}^{(k)}), \quad \Pi_*^\perp := \Pi^\perp(\boldsymbol{x}^*),
$$

where $\omega \in \{\boldsymbol{x}, \boldsymbol{u}, \alpha\}$ and evaluate $Q_k$ to obtain

$$
\begin{aligned}
Q_k &= \alpha_k \nabla_{\mathcal{M}}^2 g(\boldsymbol{x}^{(k+1)}, \boldsymbol{u}^*) + \Pi_{k+1} + \mathfrak{W}_{\boldsymbol{x}^{(k+1)}}(\cdot, \Pi_{k+1}^{\perp}(\boldsymbol{x}^{(k+1)} - G(\boldsymbol{x}^{(k)}, \boldsymbol{u}^*, \alpha_k))) \\
&= \alpha_k \nabla_{\mathcal{M}}^2 g(\boldsymbol{x}^*, \boldsymbol{u}^*) + \alpha_k \mathfrak{W}_{\boldsymbol{x}^{(k+1)}}(\cdot, \Pi_{k+1}^{\perp} \nabla_{\boldsymbol{x}} f(\boldsymbol{x}^{(k+1)}, \boldsymbol{u}^*)) + \Pi_{k+1} + \\
&\quad \mathfrak{W}_{\boldsymbol{x}^{(k+1)}}(\cdot, \Pi_{k+1}^{\perp}(G(\boldsymbol{x}^{(k+1)}, \boldsymbol{u}^*, \alpha_k) - G(\boldsymbol{x}^{(k)}, \boldsymbol{u}^*, \alpha_k))) \\
&= \alpha_k \nabla_{\mathcal{M}}^2 F(\boldsymbol{x}^*, \boldsymbol{u}^*) - \Pi_{k+1} \nabla_{\boldsymbol{x}}^2 f(\boldsymbol{x}^{(k+1)}, \boldsymbol{u}^*) \Pi_{k+1} + \Pi_{k+1} + \\
&\quad \mathfrak{W}_{\boldsymbol{x}^{(k+1)}}(\cdot, \Pi_{k+1}^{\perp}(G(\boldsymbol{x}^{(k+1)}, \boldsymbol{u}^*, \alpha_k) - G(\boldsymbol{x}^{(k)}, \boldsymbol{u}^*, \alpha_k))).
\end{aligned}
$$

Similarly, $\tilde{Q}_k$ is given by

$$
\tilde{Q}_k = \alpha_k \nabla_{\mathcal{M}}^2 F(\boldsymbol{x}^*, \boldsymbol{u}^*) - \alpha_k \Pi_* \nabla_{\boldsymbol{x}}^2 f(\boldsymbol{x}^*, \boldsymbol{u}^*) \Pi_* + \Pi_* .
$$

From Liang et al. (2017, Lemma 4.3), $\alpha_k \nabla_{\mathcal{M}}^2 F(\boldsymbol{x}^*, \boldsymbol{u}^*) - \alpha_k \Pi_* \nabla_{\boldsymbol{x}}^2 f(\boldsymbol{x}^*, \boldsymbol{u}^*) \Pi_*$ is positive semi-definite and the eigenvalues of $\tilde{Q}_k^{\dagger}$ lie in $(0, 1]$. Moreover, the non-expansiveness of $G(\cdot, \boldsymbol{u}^*, \alpha_k)$ and the continuity of $(\boldsymbol{x}, \boldsymbol{v}) \mapsto \mathfrak{W}_{\boldsymbol{x}}(\cdot, \boldsymbol{v})$ due to the $C^2$-smoothness of $\mathcal{M}$ implies that $\mathfrak{W}_{\boldsymbol{x}^{(k+1)}}(\cdot, \Pi_{k+1}^{\perp}(G(\boldsymbol{x}^{(k+1)}, \boldsymbol{u}^*, \alpha_k) - G(\boldsymbol{x}^{(k)}, \boldsymbol{u}^*, \alpha_k))) \to 0$. This entails that $Q_k - \tilde{Q}_k \to 0$ and the eigenvalues of $Q_k^{\dagger}$ are also eventually bounded. Hence, $Q_k^{\dagger} - \tilde{Q}_k^{\dagger}$, which can be rewritten as

$$
\begin{aligned}
Q_k^{\dagger} - \tilde{Q}_k^{\dagger} &= Q_k^{\dagger} - Q_k^{\dagger} \Pi_* + Q_k^{\dagger} \Pi_* - \Pi_k \tilde{Q}_k^{\dagger} + \Pi_k \tilde{Q}_k^{\dagger} - \tilde{Q}_k^{\dagger} \\
&= Q_k^{\dagger} \Pi_k - Q_k^{\dagger} \Pi_* + Q_k^{\dagger} \tilde{Q}_k \tilde{Q}_k^{\dagger} - Q_k^{\dagger} Q_k Q_*^{\dagger} + \Pi_k \tilde{Q}_k^{\dagger} - \tilde{Q}_k^{\dagger} \Pi_* \\
&= Q_k^{\dagger}(\Pi_k - \Pi_*) - Q_k^{\dagger}\left(Q_k - \tilde{Q}_k\right)\tilde{Q}_k^{\dagger} + (\Pi_k - \Pi_*)\tilde{Q}_k^{\dagger},
\end{aligned}
$$

also converges to 0. Similarly $P_{\omega,k} - \tilde{P}_{\omega,k} \to 0$ for all $\omega \in \{\boldsymbol{x}, \boldsymbol{u}, \alpha\}$ because

$$
\begin{aligned}
P_{\boldsymbol{x},k} - \tilde{P}_{\boldsymbol{x},k} &= \Pi_{k+1}\left(I - \alpha_k \nabla_{\boldsymbol{x}}^2 f(\boldsymbol{x}^{(k)}, \boldsymbol{u}^*)\right) - \Pi_*\left(I - \alpha_k \nabla_{\boldsymbol{x}}^2 f(\boldsymbol{x}^*, \boldsymbol{u}^*)\right) \\
&= (\Pi_{k+1} - \Pi_*) - \alpha_k\left(\Pi_{k+1}\nabla_{\boldsymbol{x}}^2 f(\boldsymbol{x}^{(k)}, \boldsymbol{u}^*) - \Pi_*\nabla_{\boldsymbol{x}}^2 f(\boldsymbol{x}^*, \boldsymbol{u}^*)\right) \\
P_{\boldsymbol{u},k} - \tilde{P}_{\boldsymbol{u},k} &= \alpha_k\left(D_{\boldsymbol{u}}\nabla_{\mathcal{M}} g(\boldsymbol{x}^{(k+1)}, \boldsymbol{u}^*) + D_{\boldsymbol{u}}\Pi_{k+1}\nabla_{\boldsymbol{x}} f(\boldsymbol{x}^{(k)}, \boldsymbol{u}^*) - D_{\boldsymbol{u}}\nabla_{\mathcal{M}} F(\boldsymbol{x}^*, \boldsymbol{u}^*)\right) \\
P_{\alpha,k} - \tilde{P}_{\alpha,k} &= \nabla_{\mathcal{M}} g(\boldsymbol{x}^{(k+1)}, \boldsymbol{u}^*) + \Pi_{k+1}\nabla_{\boldsymbol{x}} f(\boldsymbol{x}^{(k)}, \boldsymbol{u}^*) - \nabla_{\mathcal{M}} F(\boldsymbol{x}^*, \boldsymbol{u}^*)
\end{aligned}
$$

This concludes the proof because from (9), $D\mathcal{A}_\alpha(\boldsymbol{x}, \boldsymbol{u})$ is given by,

$$
D\mathcal{A}_\alpha(\boldsymbol{x}, \boldsymbol{u}) = -\nabla_{\mathcal{M}}^2 H(\mathcal{A}_\alpha(\boldsymbol{x}, \boldsymbol{u}), \boldsymbol{x}, \boldsymbol{u}, \alpha)^{\dagger} D_{(\boldsymbol{x}, \boldsymbol{u}, \alpha)}\nabla_{\mathcal{M}} H(\mathcal{A}_\alpha(\boldsymbol{x}, \boldsymbol{u}), \boldsymbol{x}, \boldsymbol{u}, \alpha). \tag{35}
$$

(iii) The expressions for $Q_k - \tilde{Q}_k$ and $P_{\omega,k} - \tilde{P}_{\omega,k}$ for $\omega \in \{\boldsymbol{x}, \boldsymbol{u}, \alpha\}$ clearly indicate that these sequences converge linearly under the given assumptions.

$\square$

## C.2. Proof of Theorem 3.7

*Proof.* We again set $\boldsymbol{x}^{(k)} := \boldsymbol{x}^{(k)}(\boldsymbol{u}^*)$ for simplicity. Thanks to Theorem 3.4(i), $\mathcal{A}_{\alpha_k}$ are $C^1$-smooth near $(\boldsymbol{x}^{(k)}, \boldsymbol{u}^*, \alpha_k)$ for all $k \in \mathbb{N}$ provided that $\boldsymbol{x}^{(0)}$ is sufficiently close to $\boldsymbol{x}^*$. Differentiation of the fixed-point iteration $\boldsymbol{x}^{(k+1)} := \mathcal{A}_{\alpha_k}(\boldsymbol{x}^{(k)}, \boldsymbol{u}^*)$ with respect to $\boldsymbol{u}$ yields

$$
D\boldsymbol{x}^{(k+1)}(\boldsymbol{u}^*) = D_{\boldsymbol{x}}\mathcal{A}_{\alpha_k}(\boldsymbol{x}^{(k)}, \boldsymbol{u}^*)\Pi(\boldsymbol{x}^{(k)})D\boldsymbol{x}^{(k)}(\boldsymbol{u}^*) + D_{\boldsymbol{u}}\mathcal{A}_{\alpha_k}(\boldsymbol{x}^{(k)}, \boldsymbol{u}^*), \tag{36}
$$

and Theorem 3.6 asserts that under the given assumptions, for all $k \in \mathbb{N}$,

$$
D\psi(\boldsymbol{u}^*) = D_{\boldsymbol{x}}\mathcal{A}_{\alpha_k}(\boldsymbol{x}^*, \boldsymbol{u}^*)\Pi(\boldsymbol{x}^*)D\psi(\boldsymbol{u}^*) + D_{\boldsymbol{u}}\mathcal{A}_{\alpha_k}(\boldsymbol{x}^*, \boldsymbol{u}^*). \tag{37}
$$

Just like the arguments made in the proof of Theorem A.12, for any $\dot{\boldsymbol{u}} \in \mathcal{U}$, if we define

$$
\begin{aligned}
\boldsymbol{e}^{(k)} &:= D\boldsymbol{x}^{(k)}(\boldsymbol{u}^*)\dot{\boldsymbol{u}} - D\psi(\boldsymbol{u}^*)\dot{\boldsymbol{u}} \\
B_k &:= D_{\boldsymbol{x}}\mathcal{A}_{\alpha_k}(\boldsymbol{x}^*, \boldsymbol{u}^*)\Pi(\boldsymbol{x}^*) \\
C_k &:= D_{\boldsymbol{x}}\mathcal{A}_{\alpha_k}(\boldsymbol{x}^{(k)}, \boldsymbol{u}^*)\Pi(\boldsymbol{x}^{(k)}) - D_{\boldsymbol{x}}\mathcal{A}_{\alpha_k}(\boldsymbol{x}^*, \boldsymbol{u}^*)\Pi(\boldsymbol{x}^*) \\
\boldsymbol{d}^{(k)} &:= (D\mathcal{A}_k(\boldsymbol{x}^{(k)}, \boldsymbol{u}^*) - D\mathcal{A}_k(\boldsymbol{x}^*, \boldsymbol{u}^*))(D\psi(\boldsymbol{u}^*)\dot{\boldsymbol{u}}, \dot{\boldsymbol{u}}),
\end{aligned} \tag{38}
$$

and subtract (37) from (36), we obtain the recursion $e^{(k+1)} = B_k e^{(k)} + C_k e^{(k)} + d^{(k)}$. From Theorem 3.6, the continuity of $\rho$ and $D\mathcal{A}_\alpha$, $\sup_{k\in\mathbb{N}} \|B_k\| < 1$ for all $k \in \mathbb{N}$ and from Theorem 3.4(ii), $C_k \to 0$ and $d^{(k)} \to 0$. Under the additional assumptions, $C_k$ and $d^{(k)}$ converge linearly with rate $q_x$ due to Theorem 3.4(i). Therefore, the (linear) convergence of the derivative sequence $(D x^{(k)}(u^*))_{k\in\mathbb{N}}$ follows from Theorem A.6. The error bound in (12), as well as the expressions for the constants $\xi_1$ and $\xi_2$, can be derived from (23) and (24), respectively where $C_K$, $e^{(K)}$, and $d^{(K)}$ are defined in (38). $\qquad\square$

## C.3. Proof of Theorem 3.10

*Proof.* (i) This part follows from Mehmood & Ochs (2022, Corollary 33) and the convergence of $x^{(k)}(u^*)$ and $\alpha_k$.

(ii) We set $z := (x_1, x_2)$, and $y := x_1 + \beta(x_1 - x_2)$ and write the expression for $D\mathcal{A}_{\alpha,\beta}(z,u)$ for $(z, u, \alpha, \beta) \in V_{\alpha_*} \times [0,1]$:

$$D_z \mathcal{A}_{\alpha,\beta}(z,u) = \begin{bmatrix} (1+\beta) D_x \mathcal{A}_\alpha(y,u) & -\beta D_x \mathcal{A}_\alpha(y,u) \\ I & 0 \end{bmatrix}$$

$$D_u \mathcal{A}_{\alpha,\beta}(z,u) = \begin{bmatrix} D_u \mathcal{A}_\alpha(y,u) \\ 0 \end{bmatrix}$$

$$D_\alpha \mathcal{A}_{\alpha,\beta}(z,u) = \begin{bmatrix} D_\alpha \mathcal{A}_\alpha(y,u) \\ 0 \end{bmatrix},$$

as provided in Mehmood & Ochs (2022, Corollary 33), where $D\mathcal{A}_\alpha(y,u)$ is given in (35). It is easy to see that the mapping $(z, u, \alpha, \beta) \mapsto D\mathcal{A}_{\alpha,\beta}(z,u)$ is continuous on $V_{\alpha_*} \times [0,1]$. Therefore, the sequences $(D\mathcal{A}_{\alpha_k,\beta_k}(x^{(k)}(u^*), x^{(k-1)}(u^*), u^*))_{k\geq K}$ and $(D\mathcal{A}_{\alpha_k,\beta_k}(x^*, x^*, u^*))_{k\geq K}$ both converge to $D\mathcal{A}_{\alpha_*,\beta_*}(x^*, x^*, u^*)$ and their difference converges to 0.

(iii) Because $\beta_k \in [0,1]$ and under the additional assumptions, $D\mathcal{A}_{\alpha_k}(y^{(k)}, u^*) - D\mathcal{A}_{\alpha_k}(x^*, u^*) = \mathcal{O}(y^{(k)} - x^*)$ as established in Theorem 3.4(iii), where $y^{(k)} := x^{(k)}(u^*) + \beta_k(x^{(k)}(u^*) - x^{(k-1)}(u^*))$, the result follows directly from the expressions of $D\mathcal{A}_{\alpha,\beta}$.

$\qquad\square$

## C.4. Proof of Theorem 3.12

*Proof.* From Theorem 3.10 the mapping $(x_1, x_2, u, \alpha) \mapsto \mathcal{A}_{\alpha_k,\beta_k}(x_1, x_2, u, \alpha)$ is $C^1$-smooth on $V_{\alpha_*}$. Therefore, when $x^{(0)}$ is close enough to $x^*$, we have $(x^{(k)}, x^{(k-1)}, u^*, \alpha_k) \in V_{\alpha_*}$ where $x^{(k)} := x^{(k)}(u^*)$ and we obtain the following recursion for the derivative iterates

$$\begin{bmatrix} D x^{(k+1)}(u^*) \\ D x^{(k)}(u^*) \end{bmatrix} = D_z \mathcal{A}_{\alpha_k,\beta_k}(x^{(k)}, u^*) \Pi(x^{(k)}, x^{(k-1)}) \begin{bmatrix} D x^{(k)}(u^*) \\ D x^{(k-1)}(u^*) \end{bmatrix} + D_u \mathcal{A}_{\alpha_k,\beta_k}(x^{(k)}, u^*)$$

From Theorem 3.11, we have

$$\begin{bmatrix} D\psi(u^*) \\ D\psi(u^*) \end{bmatrix} = D_z \mathcal{A}_{\alpha,\beta}(x^*, u^*) \Pi(x^*, x^*) \begin{bmatrix} D\psi(u^*) \\ D\psi(u^*) \end{bmatrix} + D_u \mathcal{A}_{\alpha,\beta}(x^*, u^*),$$

for $\beta = \beta_*$ and $\beta = \beta_k$ for all $k \in \mathbb{N}$. Therefore, for any $\dot{u} \in \mathcal{U}$, we define

$$e^{(k)} := (D x^{(k)}(u^*) - D\psi(u^*), D x^{(k-1)}(u^*) - D\psi(u^*))\dot{u}$$
$$B_k := D_x \mathcal{A}_{\alpha_k,\beta_k}(x^*, x^*, u^*)\Pi(x^*, x^*), \quad B_* := D_x \mathcal{A}_{\alpha_*,\beta_*}(x^*, x^*, u^*)\Pi(x^*, x^*)$$
$$C_k := D_x \mathcal{A}_{\alpha_k,\beta_k}(x^{(k)}, x^{(k-1)}, u^*)\Pi(x^{(k)}, x^{(k-1)}) - D_x \mathcal{A}_{\alpha_k,\beta_k}(x^*, x^*, u^*)\Pi(x^*, x^*)$$
$$d^{(k)} := (D\mathcal{A}_{\alpha_k,\beta_k}(x^{(k)}, x^{(k-1)}, u^*) - D\mathcal{A}_{\alpha_k,\beta_k}(x^*, x^*, u^*))(D\psi(u^*)\dot{u}, D\psi(u^*)\dot{u}, \dot{u})$$

and obtain the recursion $e^{(k+1)} = B_k e^{(k)} + C_k e^{(k)} + d^{(k)}$. Notice that $\rho(B_*) < 1$ from Theorem 3.11 and $B_k \to B_*$ from Theorem 3.10(i). Moreover $C_k \to 0$, and $d^{(k)} \to 0$ owing to Theorem 3.10(ii) and under the additional assumptions, the convergence of $C_k$ and $d^{(k)}$ is linear with rate $q_x$ thanks to Theorem 3.10(iii). Therefore, from Corollary A.7, the sequence $D x^{(k)}(u^*)\dot{u}$ converges (linearly) to $D\psi(u^*)\dot{u}$ for all $\dot{u} \in \mathcal{U}$. $\qquad\square$

## D. Experimental Details

In this section, we fill out some missing details from Section 4. We do not perform any feature transformation apart from batch normalization. For (16), the dataset is artificially generated with $70 \leq M \leq 90$ and $N = 200$. In particular, for each of the 50 problems, we generate $X$ and a sparse vector $\boldsymbol{w}' \in \mathbb{R}^N$ with only 50 non-zero elements. We sample each element of $X$ from standard uniform distribution and each non-zero element of $\boldsymbol{w}'$ from standard normal distribution respectively. We then generate $\boldsymbol{y}$ by computing $X\boldsymbol{w} + \boldsymbol{\varepsilon}$ where each $\boldsymbol{\varepsilon}$ is drawn from normal distibution with mean 0 and covariance matrix $10^{-3}I$. The problem has a $\rho(X^T X)$-Lipschitz continuous gradient but is not even strictly convex. However, for each choice of $\lambda \sim U(0, 10)$, $X$, and $\boldsymbol{y}$, Assumptions 3.2 and 3.3 were satisfied for each experiment. The spaces and sets mentioned in Assumption 3.1 are $\mathcal{X} = \mathbb{R}^N, \mathcal{U} = \mathbb{R}^{M \times N} \times \mathbb{R}^M \times \mathbb{R}, \Omega = \mathbb{R}^{M \times N} \times \mathbb{R}^M \times (0, +\infty)$, and $\mathcal{M} = \{\boldsymbol{w} \in \mathbb{R}^N : \operatorname{supp}(\boldsymbol{w}) \subset \operatorname{supp}(\boldsymbol{w}^*)\}$ where $\operatorname{supp}(\boldsymbol{w})$ denotes the support of $\boldsymbol{w}$. For (17), we use MADELON dataset with $M = 2,000$ samples and $N = 501$ features. For 50 set of experiments, we sample $\lambda \sim \mathcal{N}(0, 10^{-3})$ and perturb each element of $X$ with a Gaussian noise with standard deviation $10^{-3}$. The problem is $\lambda$-strongly convex and has a $\rho(X^T X)/(4M) + \lambda$-Lipschitz continuous gradient with $\mathcal{M} = \mathcal{X} = \mathbb{R}^N, \mathcal{U} = \mathbb{R}^{M \times N} \times \mathbb{R}$ and $\Omega = \mathbb{R}^{M \times N} \times (0, +\infty)$. For (18), we use CIFAR10 dataset with $M = 50,000$ samples $N = 32 \times 32 \times 3$ features. The problem is $\lambda$-strongly convex and has a $\rho(X^T X)/M + \lambda$-Lipschitz continuous gradient with $\mathcal{M} = \mathcal{X} = \mathbb{R}^{N \times C}, \mathcal{U} = \mathbb{R}^{M \times N} \times \mathbb{R}$ and $\Omega = \mathbb{R}^{M \times N} \times (0, +\infty)$.

We obtain a very good estimate of $\boldsymbol{w}^*$ for (16) by running APG for sufficiently large number of iterations, for (17) by using Newton's method with backtracking line search, and for (18) by running the Heavy-ball method with optimal parameters, also, for large number of iterations. The derivative of the solution $D\psi(\boldsymbol{u}^*)$ is computed by solving the linear system (2) for (multinomial) logistic regression and solving the reduced system (9) after identifying the support for lasso regression (Bertrand et al., 2022). For each problem and for each experiment, we run PGD with four different choices of step size, namely, (i) $\alpha_k = 2/(L + m)$ for (17) and $\alpha_k = 1/L$ for (16), (ii) $\alpha_k \sim U(0, \frac{2}{3L})$, (iii) $\alpha_k \sim U(\frac{2}{3L}, \frac{4}{3L})$ , and (iv) $\alpha_k \sim U(\frac{4}{3L}, \frac{2}{L})$, for each $k \in \mathbb{N}$. We also run APG with $\alpha_k = 1/L$ and $\beta_k = (k - 1)/(k + 5)$. Before starting each algorithm, we obtain $\boldsymbol{w}^{(0)} \in B_{10^{-2}}(\boldsymbol{w}^*)$ by partially solving each problem through APG. For computational reasons, instead of generating the sequence $D\boldsymbol{w}^{(k)}(\boldsymbol{u})$, we compute the directional derivative $D\boldsymbol{w}^{(k)}(\boldsymbol{u})\dot{\boldsymbol{u}}$. The vector $\dot{\boldsymbol{u}}$ belongs to the same space as $\boldsymbol{u}$ and is fixed for the 5 different algorithms being compared for each experiment and problem. For each problem, we plot the error sequences $\|\boldsymbol{w}^{(k)}(\boldsymbol{u}) - \psi(\boldsymbol{u})\|$ and $\|D\boldsymbol{w}^{(k)}(\boldsymbol{u})\dot{\boldsymbol{u}} - D_{\boldsymbol{u}}\psi(\boldsymbol{u})\dot{\boldsymbol{u}}\|$.

