# OpenReview forum: "Automatic Differentiation of Optimization Algorithms with Time-Varying Updates"
_ICML.cc/2025/Conference — ICML 2025 poster_

### Official Review · Reviewer_6V9C · 2025-03-06

**Overall Recommendation:** 2

**Summary:**

The paper studies the convergence of the gradient of algorithm iterates with respect to the hyperparameters in settings where optimization algorithms employ time-varying update rules, such as changing step sizes or momentum parameters. The authors provide convergence guarantees for the derivative of the iterates under more general assumptions on the update mapping. They adapt these results to proximal gradient descent with variable step sizes and its accelerated counterpart, FISTA, under partially smooth assumptions.

**Claims And Evidence:**

Please refer to "Other Strengths And Weaknesses" section.

**Essential References Not Discussed:**

It would be more understandable if the authors added more details on the results of Beck (1994) that they claim to improve upon.

**Experimental Designs Or Analyses:**

Yes.

**Methods And Evaluation Criteria:**

Please refer to "Other Strengths And Weaknesses" section.

**Other Comments Or Suggestions:**

1.Typo on line 264: if I understand correctly, it should state that the Hessian of $f$ is positive definite.
2.Typo on the 4th line of the first paragraph in Section 3.3: what is Algorithm 3.3?

**Other Strengths And Weaknesses:**

Strengths:
1. It provides convergence guarantees for the gradient of the algorithm iterates, extending its analysis in Beck (1994).
2. The assumption on the update mapping for convergence guarantees of the derivative of the iterates is more general.


Weaknesses:
1.  Too many assumptions. For example:
    (a) The key assumption, Assumption 2.2, requires the derivative of the update mapping $A_k$ to converge while $x^k$ is also approaching $x^*$. It is hard to understand how this condition can be satisfied. The sufficient condition, given as the equicontinuity of $\mathcal{A}_k$, is not well defined, and its meaning is unclear.

    (b) The assumption regarding the related interior is not supported by examples. Moreover, this assumption is difficult to satisfy. For instance, when $x \neq 0$, $\text{ri} (\partial \|x\|_1)$ is empty. Therefore, more successful examples are needed to justify this assumption.

    (c) The analysis would be more convincing if concrete examples were provided, particularly in the partly smooth setting. This is especially important in Theorem 3.4 (iii), where the function is required to be $C^3$ on $\mathcal{M}$.


2. Unclear motivation. This work briefly mentions in the introduction (lines 65--67) that the effective use of automatic differentiation (AD) relies on the convergence of the derivative of iterates. However, it would be more convincing if examples were provided to illustrate why the convergence of $Dx^k(u)$ is important. What do the authors mean by "effective use"? In bilevel optimization, for example, the convergence of $x^k(u)$ can be analyzed without considering the convergence of $Dx^k(u)$. Why, then, is it necessary to analyze the convergence of $Dx^k(u)$?

3. Difficult to read. Mathematical concepts are not well explained or supported by examples. The paper assumes familiarity with numerous optimization notations and concepts without providing definitions. For example, there is no definition of $\mathcal{L}(\mathcal{X},\mathcal{X})$, the relative interior mentioned in lines 237--238, or equicontinuity in line 166. Additionally, the meaning of "Lipschitz continuity in $x$ uniformly in $u$ and $k$" is unclear.
\end{enumerate}

**Questions For Authors:**

None.

**Relation To Broader Scientific Literature:**

I'm not sure if there is a strong relation between this paper with the broader scientific literature. Please refer to "Other Strengths And Weaknesses" section.

**Theoretical Claims:**

Please refer to "Other Strengths And Weaknesses" section.

---

> ### Author Rebuttal · Authors · 2025-04-01
>
> We thank the reviewer for their valuable comments and suggestions. We will fix minor typos and incorporate the suggested changes to the next version.
>
> > It would be more understandable if the authors added more details on the results of Beck (1994) that they claim to improve upon.
>
> In the appendix, we dedicate the entire Section A.2 to the results of Beck (1994) by presenting them in a way which makes the comparison between the two works easier. We could not do that in the main text due to the page limit.
>
> > (a) The key assumption, Assumption 2.2, requires the derivative ...
>
> Please see the answer to "Difficult to read. Mathematical concepts are not well explained..." below.
>
> > (b) The assumption regarding the related interior is not supported by examples ...
>
> The relative interior of a non-empty convex set is always non-empty (Rockafellar, 1997; Theorem 6.2). When $C$ is a singleton, $\mathrm {ri} \ C = C$. For practical non-smooth functions, this assumption (Assumption 3.3) is satisfied almost everywhere by a solution (Vaiter et al., 2017; Definition 3 & Theorem 3).
>
> > (c) The analysis would be more convincing if concrete examples were provided ...
>
> Please see our response "Assumptions 3.1 - 3.3 are quite less restrictive ..." to Reviewer QxMK for the question "I did not check the proof ..."
>
> > Unclear motivation. This work briefly mentions in the introduction ...
>
> When we apply AD on the algorithm for solving the lower level problem, we really are replacing the lower level problem with $$
> x^{(K)} (u) := \mathcal B (x^{(0)}, u; K) \,,
> $$ where, for a given $u$, $\mathcal B (x^{(0)}, u; K)$ performs $K$ iterations of the algorithm defined by the update mapping sequence $\mathcal A_k$, starting with $x^{(0)}$. To show that, by doing this, we are actually solving the original bilevel optimization (which does not depend on the algorithm being used to solve the lower level problem), the convergence of $x^{(k)} (u)$ to the solution $\psi (u)$ (that is, the forward pass computes the correct solution) won't suffice. We also require the guarantee that $D x^{(k)} (u)$ converges to the derivative of the solution mapping $D \psi (u)$ (that is, the backward pass computes the correct derivative) .
>
> > Difficult to read. Mathematical concepts are not ...
>
> Thank you for the suggestion. We will update our preliminaries and the notation section for a better understanding of our paper.
>
> The notion of Equicontinuity for a collection of functions (Rudin 1987; Definition 11.27) means that such a collection behaves "uniformly" in terms of continuity. For example, when $f$ is $C^2$-smooth and $g=0$ in Equation $\mathcal P$ in the paper, the update mapping of gradient descent with variable step size $\mathcal A_k (x, u) := x - \alpha_k \nabla_{x} f (x, u)$ has an equicontinuous derivative $D \mathcal A_k$.
>
> $\mathcal L (\mathcal X, \mathcal X)$ is the space of all linear operators from $\mathcal X$ to $\mathcal X$.
>
> Relative interior of a convex set is defined in, for example, Rockafellar (1997; p. 44).
>
> "Lipschitz continuity in $x$ uniformly in $u$ and $k$" means that the given collection of maps are Lipschitz continuous in $x$ where the Lipschitz constant is fixed for all $u$ and $k$. Formally, for all $u \in \Omega$, and $k \in \mathbb N$, there exists $L \in \mathbb R$, such that, for all $x, y \in \mathcal X$,
> $$
> \lVert D \mathcal A_k (x, u) - D \mathcal A_k (y, u) \rVert \leq L \lVert x - y \rVert \,.
> $$
> For the gradient descent example above, $D \mathcal A_k$ is Lipschitz continuous in $x$ uniformly in $u$ and $k$, when $\nabla_{x} f (\cdot, u)$ is $L$-Lipschitz continuous for all $u$.
>
> > Typo on line 264 ...
>
> Yes, that is a typo. It should be the Hessian of $f$ and not the gradient. Thank you for pointing it out.
>
> > Typo on the 4th line of the first paragraph ..."
>
> It should be Algorithm 1. Thank you for pointing it out.
>
> **References**
>
> Rudin, W., 1987. Real and complex analysis.
>
> Rockafellar, R.T., 1997. Convex analysis.
>
> Vaiter, S., Deledalle, C., Fadili, J., Peyré, G. and Dossal, C., 2017. The degrees of freedom of partly smooth regularizers.

---

> > ### Comment · Reviewer_6V9C · 2025-04-06
> >
> > I appreciate the authors' rebuttal and will keep my current score.

---

### Official Review · Reviewer_YwLr · 2025-03-09

**Overall Recommendation:** 4

**Summary:**

Automated differentiation (AD) is a little studied workhorse behind all major deep-learning frameworks (PyTorch, TensorFlow, JAX). The authors show linear convergence for forward-mode AD for algorithms, where the iteration's parameters change over time. This is a very neat theoretical result.

**Claims And Evidence:**

The theory is solid and its discussion is ok, although the authors could perhaps explain the relationship (overlap) with https://www.repository.cam.ac.uk/items/271dad6b-c165-434e-a4ae-74cac4129604 and https://arxiv.org/abs/2208.03107
 better. I would prefer to have a theorem or two less in the main body of the text, while having the material presented clearly original.

**Essential References Not Discussed:**

None.

**Experimental Designs Or Analyses:**

The empirical results are rudimentary, but illustrative.

**Methods And Evaluation Criteria:**

The empirical results are rudimentary, but illustrative.

**Other Comments Or Suggestions:**

In Algorithm 1 (Proximal Gradient with Extrapolation), conditions under which the step sizes ensure contraction are not apparent. If step sizes $\alpha_k$ oscillate too much, the contraction property may not hold.

In Theorem 3.6, equation (10) states that the inverse is assumed to exist, but it would be good to clarify the conditions under which this holds (i.e., ensuring positive definiteness).

The use of projection operators \(\Pi(x)\) and \(\Pi^{\perp}(x)\) from manifold-based optimization does not state whether these projections are orthogonal with respect to the Euclidean norm or not.

In Assumption 3.3, the relative interior may be empty. Is there a condition under which the relative interior is non-empty? Also, checking whether $0\in\partial_xF(x^*,u^*)$ holds can be difficult, especially in high dimensional problems.

Would the method generalize to multi-level optimization (https://arxiv.org/abs/2207.02849, https://proceedings.neurips.cc/paper/2021/hash/3de568f8597b94bda53149c7d7f5958c-Abstract.html, https://doi.org/10.1007/s00186-024-00852-5)?

Rather than the experiments presented, I would appreciate some discussion of the AD implemented in PyTorch etc.

**Other Strengths And Weaknesses:**

There are some minor typos, e.g. lines 142-143: one of "without" and "removed" should be dropped?

**Questions For Authors:**

Rather than the experiments presented, I would appreciate some discussion of the AD implemented in PyTorch etc. Would this be possible, please?

**Relation To Broader Scientific Literature:**

The discussion of Implicit differentiation could be less dismissive, perhaps? For deep learning, it is rather elegant.

**Theoretical Claims:**

The introduction could spell out that they consider Clarke subdifferential, rather than to say "We use the terminology of Rockafellar and Wets"?

"Partly smooth" functions are a bit neglected in the literateure. It would be great to relate the "partly smooth" functions to the better known semismooth functions and functions definable in o-minimal structures. (Partly smooth functions are a subclass of semismooth functions, which are a subclass of functions definable in o-minimal structures.

The proofs are mostly ok. In Lemma C.1,  intermediate equation should be:

$\Big\langle b(i), w - y(i)\Big\rangle &= - \|a^{(i)}\|\Big(\left\langle a^{(i)}, y(i) - \operatorname{proj}_{C_{k_i}}(y(i)) \right\rangle + \left\langle a^{(i)}, \operatorname{proj}_{C_{k_i}}(y(i)) - w \right\rangle\Big)$

$=-\frac{1}{\|a^{(i)}\|}.\|a^{(i)}\|^2+\frac{1}{\|a^{(i)}\|}\left\langle a(i), w-\operatorname{proj}_{C_{k_i}}(y(i)) \right\rangle\Big)\leq 0$

(where the second term is negative because of projection property). The inequality $\le-\frac{1}{2}\|a^{(i)}\|$ is a mistake in the paper.

In Theorem 2.7, the proof relies on Assumption 2.3. It might be beneficial to explicitly show how the linear convergence assumption is used in bounding  $\|Dx^k(u^*)-X^*\|$. Also, it is unclear how the dependency on $C_1$ and $C_2$ is computed; explicit bounds should be provided.

In Theorem 3.7, the assumption that $D_xA_{\alpha_k}(x^*,u^*)$ remains bounded over iterations is reasonable, but should be stated explicitly. Also, the result extends previous work naturally, but the proof would benefit from a more explicit rate bound rather than an asymptotic argument.

---

> ### Author Rebuttal · Authors · 2025-04-01
>
> We thank the reviewer for their valuable comments and suggestions. We will fix minor typos and incorporate the suggested changes to the next version.
>
> > ... the authors could ... Riis (2020) and Mehmood & Ochs (2022) better.
>
> The key difference presented in Section 1.2, Paragraph 4 (last on the page) is that "the sequences $\alpha_k$ and $\beta_k$ [must] converge" for their analysis and that "[they] do not explain the *linear* convergence of AD of APG". These are the two main points that we address in this paper. We will make it more intelligible in the updated version.
>
> > I would prefer to have ... material presented clearly original.
>
> We recall many results from other papers (Lemmas 3.2, and 3.9, Theorems 3.1, 3.6, and 3.11) and split our main results in Section~3 into intermediate results (Theorems 3.4, 3.7, 3.10, and 3.12), for a better understanding. We can change Theorems 3.4, 3.6, 3.10, and 3.11 into lemmas.
>
> > The introduction could spell ... Rockafellar and Wets"?
>
> We will update our preliminaries and the notation section to make the paper more readable. Thank you for the suggestion.
>
> > ... It would be great to relate ... definable in o-minimal structures...
>
> Can you please share a reference which relates partly smooth functions to semismooth functions?
>
> > ... In Lemma C.1, intermediate ... mistake in the paper.
>
> That is a typo. We will fix that. Thank you so much.
>
> > In Theorem 2.7, the proof ... bounding $\lVert D x^{(k)} (u^*) - X^*\rVert$.
>
> Initially, we were going to put Theorem A.2 in the main part of the paper. That would have allowed us to motivate the proof of Theorems 2.5 and 2.7 in the main part. However, the space constraints did not allow us to go through with it.
>
> > In Theorem 3.7, ... $D_{x} A_{\alpha_k}(x^*, u^*)$ remains bounded ... stated explicitly.
>
> You are right. Under the conditions of Lemma 3.2, this follows naturally. However, without imposing those assumptions, we need to explicitly assume that $x^{(k)}$ converges linearly to $x^*$ and $D_{x} A_{\alpha_k}(x^*, u^*)$ eventually has a norm less than 1.
>
> > Explicit rate bounds for Theorems 2.7 and 3.7
>
> Thanks a lot for the suggestion. We will provide the explicit error bounds with $C_1$ and $C_2$ clearly defined.
>
> > The discussion of Implicit differentiation could be ...
>
> Indeed, there is no doubt about the benefits and popularity of implicit differentiation in, for instance, the training of equilibrium models. However, because we focus on automatic differentiation, we chose to list the reasons where AD shines more than ID. We will revise our discussion on implicit differentitation.
>
> > In Algorithm 1 (Proximal Gradient with Extrapolation), conditions under ...
>
> The assumptions on the choice of the step size and extrapolation parameter in Algorithm 1 are made explicit in our lemmas and theorems by recalling the work of Liang et al. (2014; 2017) or other classical results wherever necessary. Algorithm 1 in itself is an abstract algorithm and is hence named "Proximal Gradient with Extrapolation" which only mentions the three important steps in the algorithm.
>
> > In Theorem 3.6, equation (10) ... (i.e., ensuring positive definiteness).
>
> In Theorem 3.6, because $\rho (D_{x} \mathcal A_{\alpha} (x^*, u^*) \Pi (x^*)) < 1$, we obtain $\rho (D_{x} \mathcal A_{\alpha} (x, u) \Pi (x)) < 1$ for all $u$ near $u^*$ and $x = \psi (u)$ from Theorem 3.5(i) which implies the invertibility of $I - D_{x}\mathcal A_{\alpha} (x, u) \Pi (x)$ from Lemma A.1.
>
> > The use of projection operators ... the Euclidean norm or not.
>
> The orthogonality is indeed with respect to the Euclidean inner product. The Riemannian metric is inheritied from the natural embedding of $\mathcal M$ in $\mathcal X$. We will make it clear in the paper.
>
> > In Assumption 3.3, the relative interior ... in high dimensional problems.
>
> The relative interior of a non-empty convex set is always non-empty (Rockafellar, 1997; Theorem 6.2).
>
> We assume that the reviewer mean $0\in \mathrm {ri} \ \partial_{x} F (x^*, u^*)$. For many practical partly smooth functions, it is satisfied almost everywhere by a solution (Vaiter et al. 2017; Definition 3 & Theorem 3).
>
> > Would the method generalize to multi-level optimization?
>
> If needed, our work can be put into use in multi-level optimization because we mainly analyze the problem at the lowest level. Bilevel optimization is only emphasized as an *application* of our results.
>
> > Rather than the experiments ... this be possible, please?
>
> We are sorry we could not comprehend the comment well. Should we provide the implementation details of our experiments in PyTorch?
>
> **References**
>
> Liang et al., 2014. Local linear convergence of forward--backward under partial smoothness.
>
> Liang et al., 2017. Activity identification and local linear convergence of forward--backward-type methods.
>
> Vaiter et al., 2017. The degrees of freedom of partly smooth regularizers.

---

### Official Review · Reviewer_QxMK · 2025-03-12

**Overall Recommendation:** 3

**Summary:**

This paper studies automatic differentiation which is widely used and fundamental in bilevel optimization. The focus of this work is on the case where the algorithm may have changing parameters at each iteration, such as step-sizes. Under this setting and some assumptions, they analyzed the convergence of the automatic differentiation scheme.

**Claims And Evidence:**

Yes.

**Essential References Not Discussed:**

Not find.

**Experimental Designs Or Analyses:**

Experiment settings are simple, but it is fine for a theoretical paper.

**Methods And Evaluation Criteria:**

This is a theoretical paper and there is no method proposed.

**Other Comments Or Suggestions:**

More discussions on the assumptions.

**Other Strengths And Weaknesses:**

This paper seems quite technical, and the results are not trivial. However, it is difficult for me to judge whether this paper made significant contributions. The main reason is the lack of discussions on Assumptions 3.1 - 3.3 (looks restrictive for me), which determines the applicability of their results.

**Questions For Authors:**

I can see that in page 2, the expression of $D x^{(k+1)}(u)$ includes three parts: $D_x A(x^{(k)}(u), u)$, $D x^{(k)}(u)$, and $D_u A(x^{(k)}(u), u)$. The convergence of $D x^{(k+1)}(u)$ requires all these three terms to converge. Which one is the most difficult part to prove the convergence?

**Relation To Broader Scientific Literature:**

No

**Theoretical Claims:**

I did not check the proof, while I have some concerns regarding the assumptions 3.1 - 3.3, which seem quite strong. Can the authors list some functions that satisfy these assumptions? Some discussions should be provided below these assumptions. For example, if the function F is strongly convex and smooth, will these assumptions hold?

---

> ### Author Rebuttal · Authors · 2025-04-01
>
> We thank the reviewer for their valuable comments and suggestions. We will incorporate the suggested changes to the next version.
>
> > I did not check the proof, while I have some concerns regarding the assumptions 3.1 - 3.3, which seem quite strong. Can the authors list some functions that satisfy these assumptions? Some discussions should be provided below these assumptions. For example, if the function F is strongly convex and smooth, will these assumptions hold? **...** it is difficult for me to judge whether this paper made significant contributions. The main reason is the lack of discussions on Assumptions 3.1 - 3.3 (looks restrictive for me), which determines the applicability of their results. **...** More discussions on the assumptions.
>
> Assumptions 3.1 - 3.3 are quite less restrictive when we compare them to the smooth setting ($g=0$) where $F = f$ is $C^2$-smooth and strongly convex. Many practical regularizers which induce some form of sparsity, satisfy Assumption 3.1. These include $\ell_1$ norm, $\ell_{2, 1}$ norm, and the nuclear norm etc. More examples can be found in, for instance, Vaiter et al. (2017; see Section 3). The obtained sparsity is such that the solution lies on a low dimensional manifold as compared to $\mathrm{dim} \ \mathcal X$ thus making it easier for Assumption 3.2 to be satisfied. Even though it is possible that this assumption may not satisfy, it is still way more general than the usual strongly convex, $C^2$-smooth, lower level objective assumption. Assumption 3.3 is satisfied almost everywhere by a solution for many practical regularizers (Vaiter et al. 2017; Definition 3 & Theorem 3). For linear convergence rates, in Theorems 3.7 and 3.12, we also assume local Lipschitz continuity of the second derivatives of $f$ and $g$ and $C^3$-smoothness of $\mathcal M$ which is also satisifed for the same problems listed above and more in Vaiter et al. (2017). In the udpated version, we will add these concrete examples for a better understanding of our assumptions.
>
> > I can see that in page 2, the expression of $D x^{(k+1)} (u)$ includes three parts: $D_x \mathcal A (x^{(k)} (u), u)$, $D x^{(k)} (u)$, and $D_u \mathcal A (x^{(k)} (u), u)$. The convergence of $D x^{(k+1)} (u)$ requires all these three terms to converge. Which one is the most difficult part to prove the convergence?
>
> $D x^{(k)} (u)$ and $D x^{(k+1)} (u)$ are elements of the same sequence $(D x^{(k)} (u))_{k\in\mathbb N}$, which is generated by Equation $\mathcal{DR}$ in a recursive manner whose convergence is desired. It is possible that even though the sequences
>
> $(D_{x} \mathcal{A} (x^{(k)} (u), u))_{k\in\mathbb{N}}$, and
>
>  $(D_u \mathcal{A} (x^{(k)} (u), u))_{k\in\mathbb{N}}$
>
> converge (which is straightforward when $\mathcal A$ is $C^1$), the sequence $(D x^{(k)} (u))_{k\in\mathbb N}$ does not converge. So in the end, it is $D x^{(k)} (u)$, whose convergence requires more effort to show.

---

### Official Review · Reviewer_n3Bk · 2025-03-20

**Overall Recommendation:** 3

**Summary:**

When, for example, solving a bilevel optimization $\min_{u\in\mathcal{U}}l(\psi(u),u)$ by gradient descent, a derivative of a solution mapping $\psi(u)$, that is $D\psi(u)$, needs to be computed. While either the implicit differentiation (ID), using chain rule, or the automatic differentiation (AD) can be used to compute $D\psi(u)$, the latter may be preferred due to its blackbox implementation. This paper particularly focuses on analyzing the AD. Regardless of the differentiation technique, one needs to iteratively update as $x^{(k+1)}(u) = \mathcal{A}_k(x^{(k)}(u),u)$ to approximate $\psi(u)$. And, during this process, the (forward mode) AD additionally updates $Dx^{(k)}(u)$ in a certain way to approximate $D\psi(u)$.

This paper's main contribution is to study the (forward mode) AD for time-varying iterative processes $\mathcal{A}_k$, such as gradient descent with line search, accelerated gradient method, and Quasi-Newton methods, in a unified way. (Beck, 1994) and other cited literatures have studied this setting, but they lack a (linear) convergence rate analysis, which is of main interest of this paper.

Section 2 is a direct extension of (Beck, 1994) that assumed the convergence of $\mathcal{A}_k$. This paper proves derivative iterate convergence without assuming convergence of $\mathcal{A}_k$. Moreover, with additional Assumption 2.1(ii), this paper was able to show a linear convergence rate. (Q: Was there a convergence rate analysis for the AD of non-time-varying update? I was not able to figure that out in the paper.) Section 3 then applies the established result in Section 2 to (extrapolated) proximal gradient with variable step size for the class of partly smooth functions. The convergence of the derivate iterates was already shown in (Mehmood & Ochs, 2022) (which does not assume convergence of $\mathcal{A}_k$), so the rate analysis is new here.

**Claims And Evidence:**

All claims are theoretically proven.

**Essential References Not Discussed:**

.

**Experimental Designs Or Analyses:**

This paper presents experiments on three classical machine learning problems. The numerical results align with the theory and seem sufficient, especially given the paper's theoretical focus.

**Methods And Evaluation Criteria:**

.

**Other Comments Or Suggestions:**

- Unlike the introduction, the abstract lacks a clear motivation (that should built upon existing limitations), and a complete picture of the paper. Regarding the latter, presenting the necessity of applying automatic differentiation to a (time-varying) iterative process, such as in bilevel optimization, could be helpful.
- Define $\mathcal{L}(\mathcal{X},\mathcal{X})$

- Contributions: Although the assumptions of (Beck, 1994) are given in the appendix and one can compare them with new assumptions, I suggest adding comments after the assumptions so that one can clearly identify contributions (i) and (ii) from (Beck, 1994) after they are introduced.
- Contibution (ii): I think Assumptions 2.2 and 2.3 should be added here.

**Other Strengths And Weaknesses:**

.

**Questions For Authors:**

- p3 left line 125: Does this mean that the analysis in (Riis, 2020) and (Mehmood & Ochs, 2022) that showed the convergence of the AD of APG cannot be applied to the AD of PGD with non-converging variable step size? If yes, then does Section 2 (especially Theorem 2.5) generalizes the result in (Riis, 2020) and (Mehmood & Ochs, 2022), so that you can provide guarantee for the AD of PGD with non-converging variable step size? I suggest explaining what is new in Section 2, in comparison to (Riis, 2020) and (Mehmood & Ochs, 2022) and particularly in terms of contribution (i).
- p4 right line 215: Could you further explain why, without such additional condition, the AD will not have a linear rate?

**Relation To Broader Scientific Literature:**

.

**Theoretical Claims:**

I was not able to check correctness of proofs.

---

> ### Author Rebuttal · Authors · 2025-04-01
>
> We thank the reviewer for their valuable comments and suggestions. We will fix minor typos and incorporate the suggested changes to the next version.
>
> > Was there a convergence rate analysis for the AD of non-time-varying update? I was not able to figure that out in the paper.
>
> We highlight them in Section 1.1. We mention these works right before introducing the time-varying algorithms in Section 1.2.
>
> > Unlike the introduction, the abstract lacks a clear motivation (that should built upon existing limitations), and a complete picture of the paper. Regarding the latter, presenting the necessity of applying automatic differentiation to a (time-varying) iterative process, such as in bilevel optimization, could be helpful.
>
> We will add the application of our work in Bilevel optimization in the updated version of the abstract. Thank you for pointing it out.
>
> > "Define $\mathcal L (\mathcal X, \mathcal X)$."
>
> $\mathcal L (\mathcal X, \mathcal X)$ is the space of all linear operators from $\mathcal X$ to $\mathcal X$. We will update our preliminaries and the notation section to make the paper more readable. Thank you for the suggestion.
>
> > "Contributions: Although the assumptions of Beck(1994) are given in the appendix and one can compare them with new assumptions, I suggest adding comments after the assumptions so that one can clearly identify contributions (i) and (ii) from Beck(1994) after they are introduced."
>
> That will definitely make our contributions more explicit. Thank you for the recommendation. We will do it in the updated version.
>
> > Contibution (ii): I think Assumptions 2.2 and 2.3 should be added here.
>
> That's a typo actually. Thank you for spotting that.
>
> > p3 left line 125: Does this mean that the analysis in (Riis, 2020) and (Mehmood & Ochs, 2022) that showed the convergence of the AD of APG cannot be applied to the AD of PGD with non-converging variable step size? If yes, then does Section 2 (especially Theorem 2.5) generalizes the result in (Riis, 2020) and (Mehmood & Ochs, 2022), so that you can provide guarantee for the AD of PGD with non-converging variable step size? I suggest explaining what is new in Section 2, in comparison to (Riis, 2020) and (Mehmood & Ochs, 2022) and particularly in terms of contribution (i).
>
> Riis (2020) and Mehmood & Ochs (2022) do not consider a general time-varying update rule. They consider APG or FISTA with converging step size and extrapolation parameter sequences by employing the analysis of Beck (1994). Their results can not be easily applied to PGD with a general non-converging step size sequence. Therefore, we provide an update on the results of Beck (1994) in Section 2 using Assumption 2.1(ii) and Assumption 2.2 and use it to provide stronger results than Riis (2020) and Mehmood & Ochs (2022) in Section 3.
>
> > p4 right line 215: Could you further explain why, without such additional condition, the AD will not have a linear rate?
>
> Actually, the correct statement should be that without such additional condition, it will be very challenging to *prove* that AD will have a linear rate.

---

### Decision · Program_Chairs · 2025-05-01

**Decision:**

Accept (poster)

**Comment:**

The main content of the paper is to study the (forward mode) AD for time-varying iterative processes, which covers various optimization schemes from the literature such as inertial acceleration schemes and quasi-Newton method. The paper proposes a unified approach of AD with convergence guarantees. Numerical experiments validate the efficiency of the proposed method. The paper presents a significant development over the existing literature, expanding the applicability of the method. During the discussion period, the concerns from the reviewers are properly addressed. The authors should modify the paper properly, including improving the readability of the paper and adding necessary discussions.